# Quantum simulation of Hawking radiation and curved spacetime with a superconducting on-chip black hole

Yun-Hao Shi [1,2,11], Run-Qiu Yang[3,11], Zhongcheng Xiang[1,11], Zi-Yong Ge [4], Hao Li [1,5], Yong-Yi Wang [1,2], Kaixuan Huang[6], Ye Tian[1], Xiaohui Song[1], Dongning Zheng [1,2,7] ✉, Kai Xu [1,2,6,7,8] ✉, Rong-Gen Cai [9] ✉ & Heng Fan [1,2,6,7,8,10] ✉

Hawking radiation is one of the quantum features of a black hole that can be understood as a quantum tunneling across the event horizon of the black hole, but it is quite difficult to directly observe the Hawking radiation of an astrophysical black hole. Here, we report a fermionic lattice-model-type realization of an analogue black hole by using a chain of 10 superconducting transmon qubits with interactions mediated by 9 transmon-type tunable couplers. The quantum walks of quasi-particle in the curved spacetime reflect the gravitational effect near the black hole, resulting in the behaviour of stimulated Hawking radiation, which is verified by the state tomography measurement of all 7 qubits outside the horizon. In addition, the dynamics of entanglement in the curved spacetime is directly measured. Our results would stimulate more interests to explore the related features of black holes using the programmable superconducting processor with tunable couplers.

In the classical picture, a particle falls into a black hole horizon and the horizon prevents the particle from turning back, then escape becomes impossible. However, taking into account quantum effects, the particle inside the black hole is doomed to gradually escape to the outside, leading to the Hawking radiation[1]. The problem is that direct observation of such a quantum effect of a real black hole is difficult in astrophysics. For a black hole with solar mass, the associated Hawking temperature is only ~$10^{-8}$ K and the corresponding radiation probability is astronomically small. Given by this, various analog systems were proposed to simulate a black hole and its physical effects in laboratories[2]. Over the past years, the theory of Hawking radiation has been tested in experiments based on various platforms engineered

with analog black holes, such as using shallow water waves[2-7], Bose-Einstein condensates (BEC)[8-12], optical metamaterials and light[13-15], etc.

On the other hand, the developments of superconducting processors enable us to simulate various intriguing problems of many-body systems, molecules, and to achieve quantum computational supremacy[16-19]. However, constructing an analog black hole on a superconducting chip is still a challenge, which requires wide-range tunable and site-dependent couplings between qubits to realize the curved spacetime[20]. Coincidentally, a recent architectural breakthrough of tunable couplers for superconducting circuit[21], which has been exploited to implement fast and high-fidelity two-qubit gates[22-25], offers an opportunity to achieve specific coupling distribution

[1]Institute of Physics, Chinese Academy of Sciences, 100190 Beijing, China. [2]School of Physical Sciences, University of Chinese Academy of Sciences, 100049 Beijing, China. [3]Center for Joint Quantum Studies and Department of Physics, School of Science, Tianjin University, 300350 Tianjin, China. [4]Theoretical Quantum Physics Laboratory, RIKEN Cluster for Pioneering Research, Wako-shi, Saitama 351-0198, Japan. [5]School of Physics, Northwest University, 710127 Xi'an, China. [6]Beijing Academy of Quantum Information Sciences, 100193 Beijing, China. [7]Songshan Lake Materials Laboratory, 523808 Dongguan, Guangdong, China. [8]CAS Center for Excellence in Topological Quantum Computation, University of Chinese Academy of Sciences, 100049 Beijing, China. [9]CAS Key Laboratory of Theoretical Physics, Institute of Theoretical Physics, Chinese Academy of Sciences, 100190 Beijing, China. [10]Hefei National Laboratory, 230088 Hefei, China. [11]These authors contributed equally: Yun-Hao Shi, Run-Qiu Yang, Zhongcheng Xiang. ✉e-mail: dzheng@iphy.ac.cn; kaixu@iphy.ac.cn; cairg@itp.ac.cn; hfan@iphy.ac.cn

analogous to the curved spacetime. We develop such a super-conducting processor integrated with a one-dimensional (1D) array of 10 qubits with interaction couplings controlled by 9 tunable couplers, see Fig. 1, which can realize both flat and curved spacetime back-grounds. The quantum walks of quasi-particle excitations of super-conducting qubits are performed to simulate the dynamics of particles in a black hole background, including dynamics of an entangled pair inside the horizon. By using multi-qubit state tomography, Hawking radiation is measured which is in agreement with theoretical predic-tion. This new constructed analog black hole then facilitates further investigations of other related problems of the black hole.

## Results

### Model and setup

To consider the effects of curved spacetime on quantum matters, we consider a (1+1)-D Dirac field, of which the Dirac equation is written as $(\hbar = c = 1)$[26,27]

$$i\gamma^a e^\mu_{(a)} \partial_\mu \psi + \frac{i}{2} \gamma^a \frac{1}{\sqrt{-g}} \partial_\mu \left(\sqrt{-g} e^\mu_{(a)}\right) \psi - m\psi = 0, \quad (1)$$

where $g$ is the determinate of $g_{\mu\nu}$, the vielbein $e^{(a)}_\mu$ satisfies the ortho-normal condition $e^{(a)}_\mu e^\nu_{(a)} = \delta^\nu_\mu$ and the $\gamma$-matrices in the two-dimensional case are chosen to be $\gamma = (\sigma_z, i\sigma_y)$. In the Eddington-Finkelstein coordinates $\{t, x\}$ and in the massless limit $m \to 0$, such a

Dirac field can be quantized into a discrete XY lattice model with site-dependent hopping couplings. The effective Hamiltonian reads (see Supplementary Information and ref. 20)

$$\hat{H} = -\sum_j \kappa_j \left(\hat{\sigma}^+_j \hat{\sigma}^-_{j+1} + \hat{\sigma}^-_j \hat{\sigma}^+_{j+1}\right) - \sum_j \mu_j \hat{\sigma}^+_j \hat{\sigma}^-_j, \quad (2)$$

where $\hat{\sigma}^+_j$ ($\hat{\sigma}^-_j$) is the raising (lowering) operator of the $j$-th qubit, $\mu_j$ denotes the on-site potential, the site-dependent coupling $\kappa_j$ takes the form $\kappa_j \approx f((j - j_h + 1/2)d)/4d$ with $d$ being the lattice constant. Here, the function $f(x)$ is related to spacetime metric, which is given in the Eddington-Finkelstein coordinates $\{t, x\}$ as $ds^2 = f(x)dt^2 - 2dtdx$ (see "Methods" section and Supplementary Information). The spatial position $x$ is discretized as $x_j = (j - j_h)d$. Since the horizon locates at $f(x_h) = 0$ with $f'(x_h) > 0$, the horizon in our analogs model is then defined at site $j = j_h$ where $f(x_h) = 0$, but the sign of $\kappa_j$ is different on its two sides of the horizon resulting in a black hole spacetime structure. One side of the horizon is considered as the interior of the black hole, while the opposite side represents the exterior of the black hole.

We perform the experiment to simulate the black hole using a superconducting processor with a chain of 10 qubits $Q_1$–$Q_{10}$, which represents the Hamiltonian (2), additionally with 9 tunable couplers interspersed between every two nearest-neighbor qubits, see Fig. 1. The effective hopping coupling $\kappa_j$ between qubits $Q_j$ and $Q_{j+1}$ can be tuned arbitrarily via programming the frequency of the corresponding

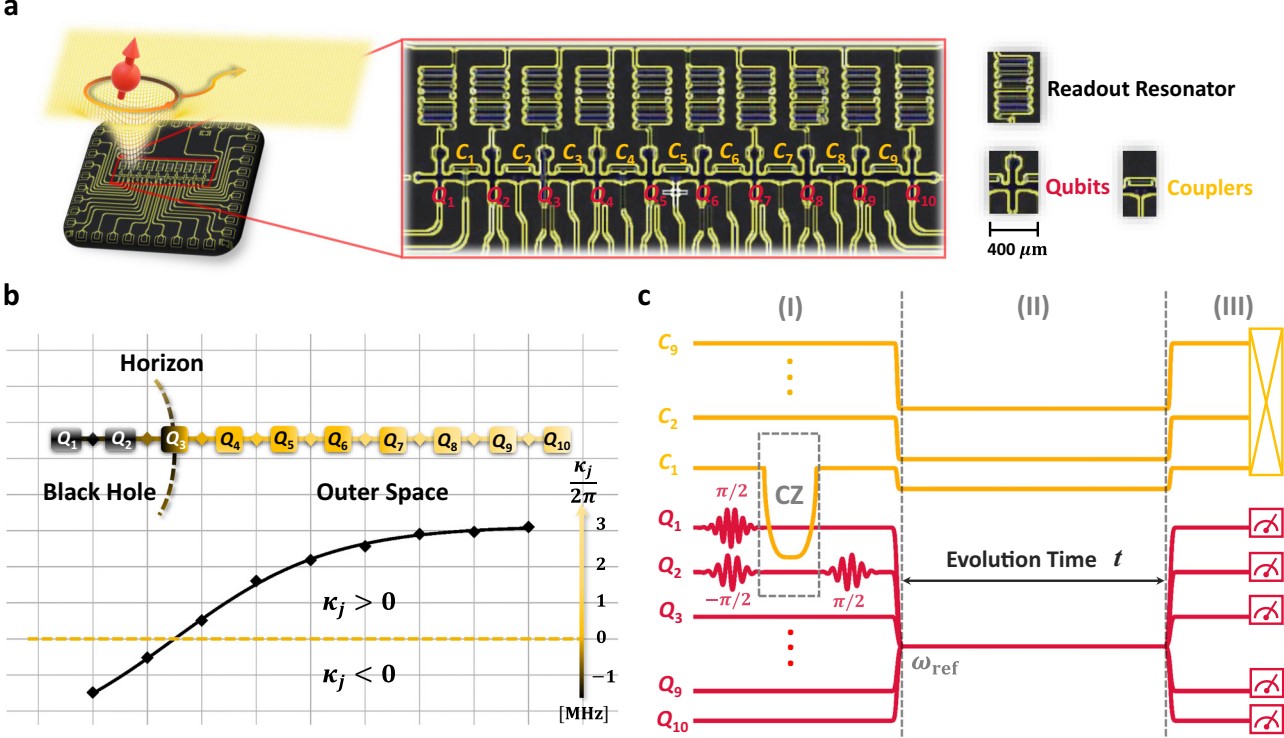

**Fig. 1 | On-chip analog black hole. a** False-color image of superconducting pro-cessor and schematic analog black hole. Ten transmon qubits, $Q_1$ - $Q_{10}$, shown as crosses, are integrated along a chain with nearest-neighbor couplings. Each nearest-neighbor two qubits are coupled via a coupler, $C_1$ - $C_9$, realized by a transmon with only a flux bias line. All the transmons are frequency-tunable, but only the qubit has the XY control line and readout resonator. The schematic image represents the background of curved spacetime simulated by this superconducting chip. The red cartoon spin located at the upper-left denotes the evolution of one quasi-particle that is initially in the black hole and the outward-going radiation. **b** Schematic representation of the site-dependent effective coupling strengths $\kappa_j$. In the experiment, the coupling $\kappa_j$ is designed according to Eq. (3). There is a boundary

analogous to the event horizon of a black hole, where the coupling changes its sign at site $Q_3$. Thus qubits $Q_1$ and $Q_2$ can be considered as the interior of the black hole, $Q_3$ is at the horizon, and $Q_4$–$Q_{10}$ are in the outside black hole. **c** Experimental pulse sequence for observing dynamics of entanglement, which consists of three parts, i.e., (I) initialization, (II) evolution, and (III) measurement. For the initialization (I), we prepare an entangled Bell pair on $Q_1Q_2$ by combining several single-qubit pulses and a two-qubit control-phase (CZ) gate. At the left boundary of region (II), the curved (or flat) spacetime forms. Then the system will evolve according to the corresponding $\kappa_j$ in the Hamiltonian for a time $t$. In region (III), we perform the state tomography measurement.

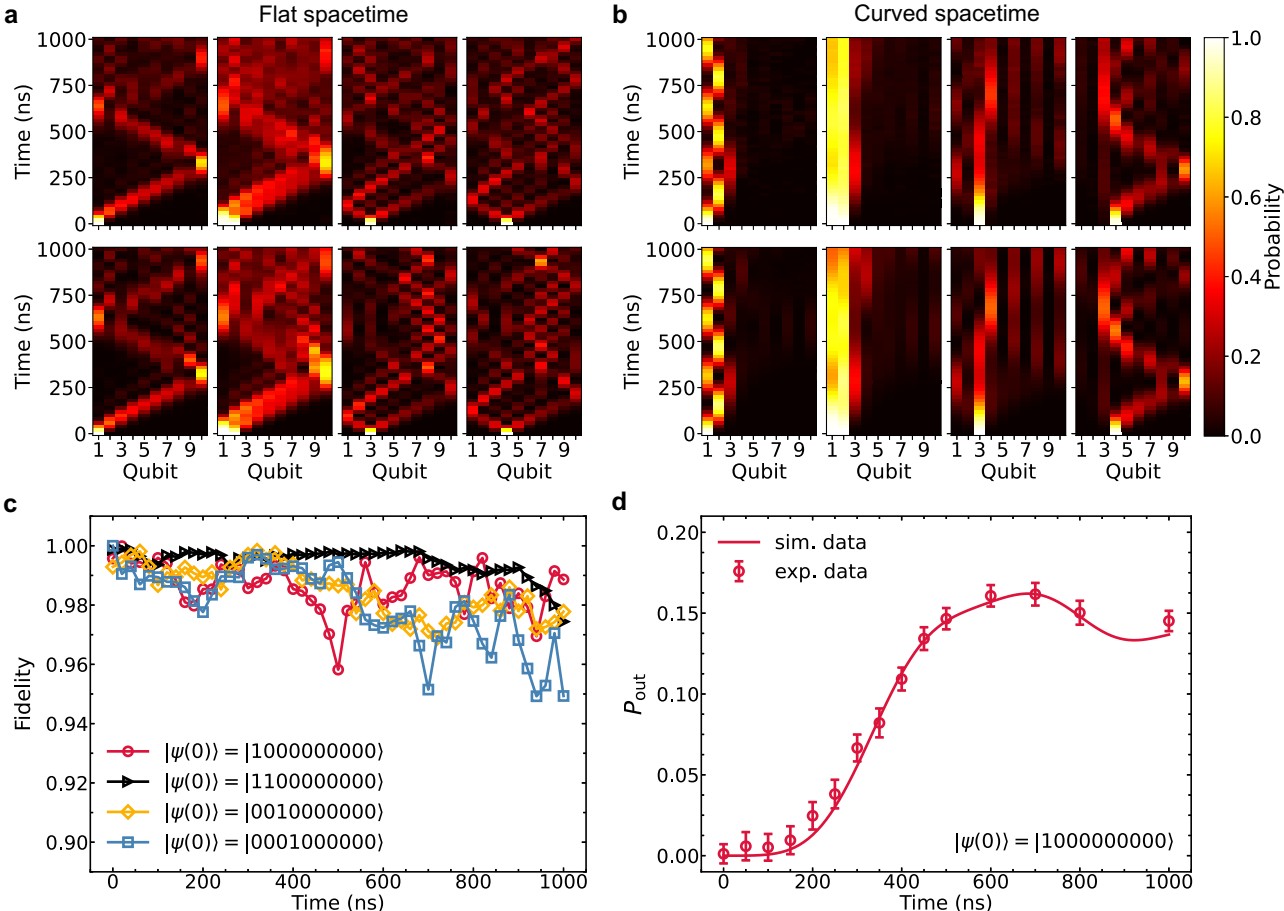

**Fig. 2 | Quantum walks in a 1D array of 10 superconducting qubits. a** Results of the quantum walks in a flat spacetime for four different initial states, i.e., $|\psi(0)\rangle = |1000000000\rangle$, $|1100000000\rangle$, $|0010000000\rangle$ and $|0001000000\rangle$ with $|0\rangle$ representing the ground state of a qubit and $|1\rangle$ the excited state. The case of black hole spacetime is presented in **b**. The heatmap denotes the probabilities of excited-state for $Q_i$ in time. The horizontal axis is indexed as qubit number $i$, the vertical axis is the evolution time. Here we show both the numerical simulation and experimental data to compare the difference between the flat spacetime ($\kappa_j/(2\pi) \approx 2.94$ MHz) and the curved spacetime ($\beta/(2\pi) \approx 4.39$ MHz). **c** The fidelity of quantum walks in the curved spacetime. **d** The probability of finding a particle outside the horizon on qubits $\{Q_4 Q_5 Q_6 Q_7 Q_8 Q_9 Q_{10}\}$. Error bars are 1 SD calculated from all probability data of 50 repetitive experimental runs.

coupler $C_j$, see "Methods" section. To describe the curved spacetime experimentally, we adjust the frequencies of all the couplers, and design the effective coupling distribution as

$$\kappa_j = \frac{\beta \tanh((j - j_h + 1/2)\eta d)}{4\eta d} \quad (3)$$

with $j_h = 3$, $\eta d = 0.35$, and $\beta/(2\pi) \approx 4.39$ MHz. Here we choose $f(x) = \beta \tanh(\eta x)/\eta$, where $\eta$ controls the scale of variation of $f$ over each lattice site, which has the dimension of $1/d$. One can verify that this function $f(x)$ gives us a nonzero Riemannian curvature tensor and so describes a 2-dimensional curved spacetime. As shown in Fig. 1b, the coupling $\kappa_j$ goes monotonically from negative to positive from $Q_3$'s left to right side. In this way, the information of the static curved spacetime background is encoded into the site-dependent coupling distribution. Thus, the site $Q_3$ where the sign of the coupling reverses can be analogous to the event horizon of the black hole, the side of negative coupling ($Q_1$-$Q_2$) can be considered as the interior of the black hole, and $Q_4$–$Q_{10}$ are outside the horizon. For comparison, we also realize a uniform coupling distribution with $\kappa_j/(2\pi) \approx 2.94$ MHz to realize a flat spacetime. In fact, from the viewpoint of the lattice qubit model, the results will be equivalent if the function $\kappa$ is replaced by $|\kappa|$ both in the case of curved and flat spacetime. Since we here map the coupling to the components of metric, the continuity requires $\kappa$ changes the sign when passing through the analog horizon.

In the experiment, we first prepare an initial state $|\psi(0)\rangle$ with quasi-particle excitations, i.e., exciting qubits or creating an entangled pair. The evolution of the initial state known as quantum walk will be governed by Schrödinger equation $|\psi(t)\rangle = e^{-iHt}|\psi(0)\rangle$ based on 1D programmable controlled Hamiltonian (2). The dynamics of the prepared states then simulate the behavior of quasi-particle in the studied (1+1)-dimensional spacetime with a designed flat or curved structure.

### Quantum walks in analog curved spacetime

Figure 2 a and b show the propagation of quasi-particles in flat and curved spacetimes, respectively. Here we initialize the system by preparing four different single-particle or two-particle states, including $|\psi(0)\rangle = |1000000000\rangle$, $|1100000000\rangle$, $|0010000000\rangle$, and $|0001000000\rangle$ with $|0\rangle$ and $|1\rangle$ being the eigenstates of $\hat{\sigma}_j^+ \hat{\sigma}_j^-$. Once the initial state is prepared, we apply the rectangular Z pulses on all qubits to quench them in resonance at a reference frequency of $\omega_{ref}/(2\pi) \approx 5.1$ GHz. Meanwhile, the hopping coupling $\kappa_j$ between qubits is fixed as Eq. (3) (curved spacetime) or a constant (flat spacetime) by controlling couplers. After evolving for time $t$, all qubits are biased back to idle points for readout. The occupation of quasi-particle density distribution $p_j(t) := \langle \psi(t)|\hat{\sigma}_j^+ \hat{\sigma}_j^-|\psi(t)\rangle$ is measured by averaging 5000 repeated single-shot measurements, as shown in Fig. 2a, b.

Figure 2a shows that the propagation of quasi-particle in the flat spacetime is unimpeded, corresponding to the result of conventional quantum walk with diffusive expansion[28–31]. In contrast, the particle is

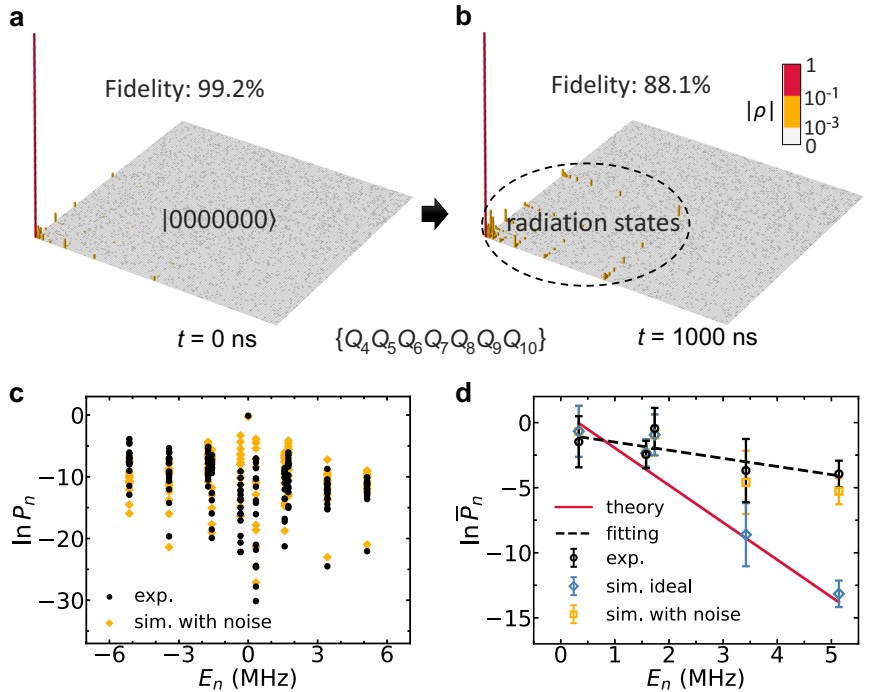

**Fig. 3 | Observation of analog Hawking radiation. a** The 7-qubit density matrix at $t = 0$ ns. Initially, only $Q_1$ is prepared in $|1\rangle$ and all the qubits outside the horizon are almost in $|0\rangle$. **b** The 7-qubit density matrix at $t = 1000$ ns after the quench dynamics. Due to the Hawking radiation, radiation states can be detected with small probabilities. The fidelity between ideal and experimental density matrix at $t = 0$ and 1000 ns are 99.2% and 88.1%, respectively. **c** The logarithmic probability of finding a particle outside the horizon $P_n$ vs. its energy $E_n$. **d** The logarithm of average radiation probability vs. the energy of particle when $E_n > 0$. Error bars are 1 SD calculated from the tomography data at the same energy. The slope of the red line represents the reciprocal of Hawking temperature without noise, where the Hawking temperature here is given by $T_H/(2\pi) = \beta/(8\pi^2) \approx 0.35$ MHz or $\approx 1.7 \times 10^{-5}$ K in Kelvin temperature. The experimental results are in agreement with the simulated data for low energy but diverge at high energy due to experiment noises.

mainly trapped in our on-chip black hole due to the analog gravity around the horizon $Q_3$, as shown in Fig. 2b with the initial state $|\psi(0)\rangle = |1000000000\rangle$ and $|\psi(0)\rangle = |1100000000\rangle$. Due to the infalling Eddington-Finkelstein coordinates we took, our model only simulates the outgoing modes of the particle (see Supplementary Information). Hence, the interior and exterior of black hole are equivalent so that the same phenomenon can be observed in that case where the particle is initially prepared in the exterior of black hole ($|\psi(0)\rangle = |0001000000\rangle$).

Here, we also present the result of the particle initialized at the horizon in Fig. 2b, i.e., $|\psi(0)\rangle = |0010000000\rangle$. In the continuous curved spacetime, the particle initialized at the horizon is bound to the horizon forever due to the zero couplings on both sides of the horizon. However, in the finite-size lattice, the coupling strengths on both sides of the horizon are not strictly zero even though they are very small ($\approx 0.54$ MHz). The particle seems to be localized at the horizon for a very short time, but it is doomed to escape from the constraints due to the finite-size effects.

To show the accuracy of the experimental results of quantum walk in the curved spacetime, we present the fidelity $F(t) = \sum_{j=1}^{10} \sqrt{p_j(t)q_j(t)}$ between the measured and theoretical probability distributions $p_j(t)$ and $q_j(t)$ in Fig. 2c. The high fidelity, greater than 97% within 400 ns experiment time, implies that our experimental results are consistent with the theoretical predictions as also demonstrated by the similarity between experimental data and numerical simulations. Note that in both cases of the flat and curved spacetimes the particle will be reflected when it arrives at the boundary $Q_1$ or $Q_{10}$.

## Observation of analog Hawking radiation

Black holes emit thermal radiation leading to evaporation, known as Hawking radiation. However, its observation is a challenge even for an analog black hole due to the accuracy of the experiment. The Hawking radiation of a black hole is spontaneous in nature. The first realization of spontaneous Hawking radiation in an analog experiment was in BEC system[8]. Here we report an observation of analog Hawking radiation on the superconducting quantum chip, which is also the first quantum realization of "lattice black hole" originally proposed by T. Jacobson more than twenty years ago[32,33].

For the initial state with a particle inside the horizon in our experiment, the evolution of the state shows the propagation of particle that results in a nonzero density of state outside the horizon is equivalent to the Hawking radiation of the black hole. Note that the Hawking radiation observed here is stimulated because we induce an excitation by flipping a qubit in $|1\rangle$.

Defining the probability of finding a particle outside the horizon as $P_{out} = \sum_{j=4}^{10} p_j$, Fig. 2d shows a rising tendency of $P_{out}$ in time. This result can be considered as an important signature of Hawking radiation for the analog black hole[3,4,14,34].

The theory of Hawking radiation points out that the probability of radiation satisfies a canonical blackbody spectrum,

$$P_{out}(E) \propto e^{-\frac{E}{T_H}}, \qquad (4)$$

where $E$ denotes the energy of particle outside the horizon, $T_H/(2\pi) = g_h/(4\pi^2)$ is defined as the effective temperature of the Hawking radiation, and $g_h = \frac{1}{2}f'(x_h) = \beta/2$ represents the surface gravity of the black hole[20]. The derivation of Eq. (4) can be constructed by using the picture of quantum tunneling to obtain the tunneling rate of particle[35–37]. We use this picture in this work to understand Hawking radiation. Such a picture is equivalent to a field theoretical viewpoint of "particle-antiparticle pairs" created around the horizon: the antiparticle (negative energy) falls into the black hole and annihilates with this particle inside the black hole, the particle outside the horizon is materialized and escapes into infinity (see Supplementary

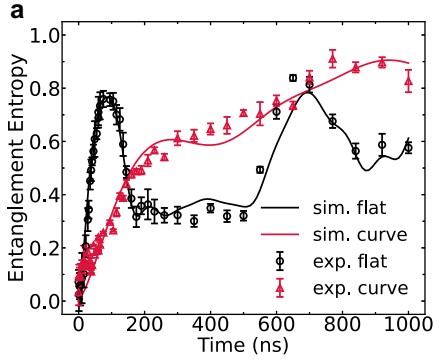
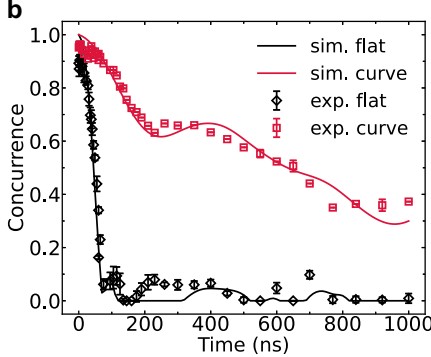

**Fig. 4 | Dynamics of entanglement in the analog black hole. a** The entanglement entropy vs. evolution time in different spacetimes. The entanglement entropy gradually increases with time in the curved spacetime. **b** The concurrence (entanglement between the pair in black hole) vs. evolution time in different spacetimes. Error bars are 1 SD calculated from all tomography data of 10 repetitive experimental runs. The rapid decline of concurrence in the flat spacetime is observed, while the concurrence in the curved spacetime is protected due to the analog gravity. The solid lines are the results of numerical simulation. Here we set the coupling for the flat spacetime to be a constant ($\kappa/(2\pi) \approx 2.94$ MHz) and $\beta/(2\pi) \approx 4.39$ MHz for the curved spacetime.

Information). Also, Eq. (4) can be viewed as the detailed balance relation between the creation and annihilation of particle around the horizon in a thermal environment[38,39].

The tunneling picture of Hawking radiation here is similar to the quantum fluid model of analog horizon[40] with two differences in details. The first one is that the analog horizon of ref. [40] is created by transonic flow but we here create analog horizon by inhomogeneous lattice hopping. The second is that the injected beam of[40] is from the subsonic region (outside horizon) so that the reflected flow stands for the flow of Hawking radiation (classically the infalling beam should be swallowed by horizon completely and there is no reflected mode), but we here create a particle inside the horizon so the transmission flow is the Hawking radiation.

To obtain the radiation probabilities, we perform the quantum state tomography (QST) on the 7 qubits ($Q_4-Q_{10}$) outside the horizon at $t = 0$ and $t = 1000$ ns, such a final time is long enough so that the particle inside the black hole has finished its tunneling to the outside but the boundary effect is negligible to the results. Here, the initial state is $|\psi(0)\rangle = |1000000000\rangle$, i.e., a particle in the black hole has a certain position. When $t = 0$ ns, no radiation can be detected and all the qubits outside the horizon are almost in $|0\rangle$, see Fig. 3a. After a long time $t = 1000$ ns, one may have a small chance to probe the particle outside the horizon, see Fig. 3b. The corresponding probabilities of radiation can be extracted from the measured 7-qubit density matrix. Assuming that $|E_n\rangle$ is the $n$-th eigenenergy of total Hamiltonian and $\hat{\rho}_{\text{out}}$ is the density matrix outside obtained by QST, then the probability of finding a particle of energy $E_n$ outside the horizon can be obtained as $P_n = \langle E_n|\hat{\rho}_{\text{out}}|E_n\rangle$, see "Methods" section. Although there are $2^{10} = 1024$ eigenstates for 10-qubit Hamiltonian Eq. (2) and the same number of $P_n$, the radiation states involve only 10 single-particle excited eigenstates due to the particle number conservation. As a consequence, only those $P_n$ that are corresponding to single-particle excited eigenstates have non-zero values, as shown in Fig. 3c. Therefore, we take the average of $P_n$ with the same positive energy $E_n$ as $\bar{P}_n$ to describe the average probability of finding a particle outside with $E_n > 0$. It can be expected that the relation between $\bar{P}_n$ and $E_n$ will agree with the theoretical prediction in Eq. (4). In Fig. 3d, the simulated results show that the logarithm of the average radiation probability is approximately linear in energy with Hawking temperature $1.7 \times 10^{-5}$ K. The fitted Hawking temperature of experimental data is around ~$7.7 \times 10^{-5}$ K, showing validity with the same order of magnitude. The deviation between experimental data and ideal simulation data is mainly caused by the evolution of the imperfect initial state. The fidelity between the imperfect initial state in the experiment and the ideal initial state is 99.2%, which may derive from the experimental noises including XY

crosstalk, thermal excitation, leakage, etc. We substitute such an experimental state for the ideal initial state in the numerical simulation of Hawking radiation, then the results of numerical simulation agree with experimental results better.

Since the analog Hawking radiation is characterized by the temperature, we then give an estimation of how large a black hole in our real universe could reproduce the same temperature. If we consider a Schwarzschild black hole in four-dimensional spacetime with the same Hawking temperature $T_H$, its mass can be calculated by $M/M_s = 6.4 \times 10^{-8}$K$/T_H$[1], where $M_s \approx 2 \times 10^{30}$ kg is the solar mass. For the simulated black hole in our work, $M/M_s \sim 10^{-3}$, whereas the typical value reported in BEC system for this quantity can be ~$10^{212}$. This significant difference in magnitude is attributed to the scales of the setup in different experimental systems. In superconducting qubits, the coupling strength is usually on the order of MHz and thus the analog surface gravity $g_h$ is of the same magnitude, leading to $T_H = g_h/(2\pi) \sim 10^{-5}$ K. Differently, the effective Hawking temperature of sonic black hole depends on the gradient of velocity at the analog horizon. The BEC system and the shallow water wave system typically give us $T_H \sim 10^{-10}$ K[9-12] and $10^{-12}$ K[4], respectively.

## Dynamics of an entangled pair in the analog black hole

Hawking predicted that the entanglement entropy increases when a black hole forms and evaporates due to the Hawking radiation. Each Hawking particle is entangled with a partner particle in the black hole. Such kind of quantum feature plays a crucial role in studying black holes and quantum information[41].

To investigate the dynamical entanglement and non-local correlation both in flat spacetime and curved spacetime, we initially prepare an entangled pair $|\psi_{\text{in}}(0)\rangle = (|00\rangle + |11\rangle)/\sqrt{2}$ (Fig. 1c). The mean fidelity of prepared entangled state is up to 99.15%. The dynamics of such an initial entangled state in flat or curved spacetime is observed by time-dependent QST measurement. We obtain the two-qubit density matrix $\hat{\rho}_{\text{in}}(t)$ from the results of QST, and use it to compute the entanglement entropy and the concurrence (see Methods). In Fig. 4a, the entanglement entropy in the case of curved spacetime progressively increases due to the Hawking radiation, while in the flat spacetime it has two wavefronts resulting from the quantum interference and reflection respectively[30]. On the other hand, the concurrence decreases with time in both cases, reflecting the process of entanglement being lost into the environment. However, the speed of entanglement propagation is limited by the gravitational effects near the horizon, and thus the decrease in concurrence is slowed in the curved spacetime case compared to the flat spacetime case, as shown in Fig. 4b.

## Discussion

In summary, we have experimentally simulated a curved spacetime of black hole and observed an analogy of Hawking radiation in a superconducting processor with tunable couplers. An high-fidelity entangled pair is also prepared inside the horizon and the corresponding dynamics of entanglement is observed. Our results may stimulate more interests to explore the related features of black holes by using programmable superconducting processor with tunable couplers, and the techniques of calibrating and controlling coupler devices will pave the way for simulating intriguing physics with quantum many-body systems of different coupling distribution.

Our current results are a step in the direction of creating quantum systems with properties analogous to those of black holes. However, many more problems remain to be addressed in a complete simulation of quantum field theory in curved spacetime, both in theory and experiment. Theoretically, it is necessary to study different dimensional systems and investigate a comprehensive theory for mapping the various gravity fields into experimental realizable models. Experimentally, it is expected to expand the category of simulated Hamiltonians, extend the scale of qubits and enhance the control accuracy. In addition to pure analog experiments, hybrid digital-analog devices with substantial flexibility in near-term applications also need to be focused[42]. Last but not the least, we must return to the basic problems of quantum field theory and try to translate more fundamental questions, for example, how generic is the emergence of gravity or what happens to spacetime when quantum corrections are fairly important[41].

## Methods

### Metric of two-dimensional spacetime

Consider a general two-dimensional spacetime background with a fixed static metric $g_{\mu\nu}$, the metric in the Schwarzschild coordinates $(t_s, x)$ reads $ds^2 = f(x)dt_s^2 - f^{-1}(x)dx^2$. To describe a black hole with nonzero temperature in 2-dimensional spacetime, we require that the function $f$ has a root at $x = x_h$ with $f'(x_h) > 0$ and $f(x) > 0$ for $x > x_h$ standing for the exterior of the black hole, while $f(x) < 0$ for $x < x_h$ for the interior. The horizon of black hole then locates at $x = x_h$. For our purpose and experimental setups, we transform above metric into "advanced Eddington-Finkelstein coordinates" by the coordinates transformation $t = t_s + \int f^{-1}(x)dx$. In the coordinates $\{t, x\}$, the metric now becomes $ds^2 = f(x)dt^2 - 2dtdx$. The differences between the "time-orthogonal coordinates" and "advanced Eddington-Finkelstein coordinates" can be found in Supplementary Note 1.

### Tunable effective couplings

To construct both flat and curved spacetime background on a single superconducting quantum chip, we use tunable coupler device. The effective coupling between nearest-neighbor qubits derives from their direct capacitive coupling and the indirect virtual exchange coupling via the coupler in between, where the former is untunable and the latter depends on the frequency of the coupler, see Supplementary Note 3. To achieve accurate control of couplings, we develop an efficient and automatic calibration for multi-qubit devices with tunable couplers, see Supplementary Note 6. In the experiments, we apply fast flux-bias Z pulses on the couplers to adjust their frequencies, contributing to the effective coupling distribution. The site-dependent coupling distribution $\kappa_j$ as Eq. (3) corresponds to the curved spacetime ($\beta/(2\pi) \approx 4.39$ MHz, Fig. 1b), while a uniform coupling distribution ($\kappa_j/(2\pi) \approx 2.94$ MHz) is related to the flat spacetime.

### Calculation of radiation probabilities

We perform the 7-qubit QST in the observation of analog Hawking radiation and obtain the density matrix outside the horizon in the 7-qubit Hilbert space. Then we set the states of the other three qubits to $|0\rangle$ and construct the density matrix in the 10-qubit Hilbert space $\hat{\rho}_{out}$. The probability of finding a particle of energy $E_n$ outside the horizon $P_n$ is calculated as $P_n = \langle E_n | \hat{\rho}_{out} | E_n \rangle$.

### Measurement of entanglement

As shown in Fig. 1c, we prepare the initial entangled pair in the black hole by using two parallel rotations $\hat{R}_{\pi/2}^y$ and $\hat{R}_{-\pi/2}^y$, a CZ gate ($\hat{U}_{CZ} = \text{diag}(1,1,1,-1)$) and a single-qubit rotation $\hat{R}_{\pi/2}^y$ in sequence. The ideal initial state of the two qubits before the quench dynamics thus is $|\psi_{in}(0)\rangle = (\hat{I} \otimes \hat{R}_{\pi/2}^y)\hat{U}_{CZ}(\hat{R}_{\pi/2}^y \otimes \hat{R}_{-\pi/2}^y)|00\rangle = (|00\rangle + |11\rangle)/\sqrt{2}$.

The state of the total system (the interior of black hole and the rest) is always a pure state during the quench dynamics. Thus, the entanglement entropy of the subsystems: $S(\hat{\rho}_{in}) = S(\hat{\rho}_{rest})$, which quantifies the entanglement contained in this bipartite quantum system. In our experiment, the cost of measuring $\hat{\rho}_{rest}$ is much higher than measuring $\hat{\rho}_{in}$ due to the dimension of the Hilbert space. Therefore, we measure $\hat{\rho}_{in}$ and calculate $S(\hat{\rho}_{in})$ as the entanglement measure.

To characteristic the entanglement between the two qubits in the black hole, we use the well-defined measure: concurrence[43], which can be calculated as $E(\hat{\rho}_{in}) = \max\{0, \lambda_1 - \lambda_2 - \lambda_3 - \lambda_4\}$ with $\lambda_i$ being the square roots of the eigenvalues of matrix $\hat{\rho}_{in}\tilde{\rho}_{in}$ in decreasing order, where $\tilde{\rho}_{in} = (\hat{\sigma}_y \otimes \hat{\sigma}_y)\hat{\rho}_{in}^*(\hat{\sigma}_y \otimes \hat{\sigma}_y)$ is the spin-flipped state of $\hat{\rho}_{in}$ with $\sigma_y$ being Pauli matrix.

## Data availability

The source data underlying all figures are available at https://doi.org/10.6084/m9.figshare.22802981. Other data are available from the corresponding author upon request.

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

## Acknowledgements

This work was supported by the Synergetic Extreme Condition User Facility (SECUF). Devices were made at the Nanofabrication Facilities at Institute of Physics in Beijing. Z.X., D.Z., K.X., and H.F. are supported by the National Natural Science Foundation of China (Grant Nos. 92265207, 92065114, T2121001, 11934018, and 11904393), Innovation Program for Quantum Science and Technology (No. 2-6), the Key-Area Research and Development Program of Guangdong Province, China (Grant No. 2020B0303030001), the Strategic Priority Research Program of Chinese Academy of Sciences (Grant No. XDB28000000), Scientific Instrument Developing Project of Chinese Academy of Sciences (Grant No. YJKYYQ20200041), and Beijing Natural Science Foundation (Grant No. Z200009). R.-Q.Y. acknowledges the support of the National Natural Science Foundation of China (Grant No. 12005155).

## Author contributions

Y.-H.S. performed the experiment with the assistance of K.X.; R.-G.C. and H.F. conceived the idea; R.-Q.Y. and R.-G.C. provided theoretical support; K.X., Y.-H.S., R.-Q.Y., Z.-Y.G. and Y.-Y.W. performed the numerical simulation; Z.X. fabricated the device with the help of D.Z. and X.S.; Y.-H.S., H.L. and K.H. helped the experimental setup supervised by K.X. with the assistance of Y.T.; H.F. supervised the whole project; Y.-H.S., R.-Q.Y., Y.-Y.W., Z.-Y.G., R.-G.C. and H.F. cowrote the manuscript, and all authors contributed to the discussions of the results and and development of the manuscript.

## Competing interests

The authors declare no competing interests.
