## [Peer Review File · Nature Communications]

REVIEWER COMMENTS

Reviewer #1 (Remarks to the Author):

I appreciate the authors' efforts to revise the manuscript in response of my and the other Referees' reports. Many technical points have been duly addressed and the manuscript has been purged by many (but not all, see below) flagrantly overselling statements.

Nonetheless, I am still not fully convinced about the scientific interest of a quantum simulation experiment where all complexities and the rich physics of the original problem are removed by complex mathematical manipulations (and quite strong hidden approximations) and one is left with a quite trivial single-particle propagation problem. In such an approach, what is interesting is the theory rather than the experiment. At odd with the authors' point of view, I still see a marked difference between the indirect nature of their observation of Hawking effect and the relatively direct evidence of Steinhauer's ones, with the consequent ease of addressing more subtle effects in this latter class of experiments.

As I already mentioned in my first report, also the entanglement features studied here are mostly reduced via a Jordan-Wigner transformation to a free-fermion model, so I strongly doubt how such an experiment can capture much of the interesting entanglement features of a real BH. Unless the authors convince me that their platform can really help understanding the question raised by [34,37-39] or "the relationship between entropy and BH horizon area", I find the text in their conclusions as just empty marketing sentences.

To summarize, I am very hesitant about my recommendation for Nat.Comms. On one hand, I agree that the experiment is a clean and solid realization on a quantum simulator of a very simple model that may be connected via complicate maths to problems of actual research interest such as Hawking radiation. On the other hand, given how strongly the experiment relies on the preliminary mathematical manipulations, I can hardly see what further developments can arise from it in the direction of analog models.

Should the editor take a positive decision, I strongly recommend the authors to make a further willing effort to put the manuscript in a correct perspective with other analog models and down-tune the remaining overselling statements, as well to take the following comments into due consideration.

More technical remarks:

-Triggered by Referee 3's remark #6, I realized that the change in sign of κ plays no role at all, as it can be straightforwardly reverted to a positive κ (as in fig.11a) by a gauge-like transformation on the outside qu-bits, by alternatively rotating the spin state of the Q1,Q2 qubits by 0 or by π around z. This must be clear in the main text, otherwise readers may get misled.

-Triggered again by Referee 3's remark #7, I would recommend the authors to include a discussion of the "continuum limit" which, in my opinion, is much more delicate than the finite-size effects of Sec.VII-C of the SM. In particular, both the tunneling effect and the localization on the horizon may be significantly modified when a finer grid of qubits is used to simulate the QFT.

-The authors may comment the importance of the $(-i)^j$ factor in the definition of \tilde{c}_j between eq.(34) and (35) of the SM. This specific change of variables (together with the assumption of slowly varying c 's that is implicit in the continuum limit) is crucial to recover the correct form of the linear dispersion in the wave-equation (37). The authors should discuss these assumptions in full detail.

-It would be worth that the authors compare their model to the tunneling model developed in the analog model context in Phys. Rev. Lett. 94, 061302 (2005). This may be the closest atomic analog of their work.

Reviewer #3 (Remarks to the Author):

The authors have reviewed their manuscript based on the reviewers' comments and resubmitted it to Nature Communications. I am very impressed with the level of detail shown by the authors in addressing my and the other reviewers' comments, and in my opinion, the manuscript is ready to be published in Nature Communications.

Reviewer #4 (Remarks to the Author):

In this work, the authors claim to have built up experimentally an analogue of a black hole, including the detection of analogue Hawking radiation. They use an array of 10 superconducting transmon qubits which are coupled by 9 additional transmons operated as tunable couplers. The authors exploit an analogy between this system in the continuum limit -under strong dispersive conditions- and the dynamics of a Dirac field in a particular metric under particular coordinates - which the authors claim to be a curved-spacetime black-hole metric. The value of the couplings determine the existence of an horizon as a point where the coupling is 0 and have opposite signs at both sides of it. Analogue Hawking radiation would appear in the form of excited states of the qubit chain.

Unfortunately, there are several issues which prevent the interpretation of the results in the form claimed by the authors. Some of them -like the particular choice of coordinates needed- have already been discussed by previous referees. However, I believe that there are even more important issues.

- Curved spacetime: In standard Einstein gravity, it is a well-known fact that all 1+1 D spacetimes are conformally flat. Therefore, there are no 1+1 D curved spacetimes. There are two alternatives to this fact: a) one can consider that the 1+1 D spacetime is just a section of a 3+1 D spacetime (see, for instance, NJP 20 053028 (2018)). This is not what the authors are doing, since their spatial coordinate is not a radial coordinate -it takes negative values inside the horizon. b) one can consider modifications of Einsteinian gravity, as in the papers of Mann and collaborators mentioned by the authors. Only in this non-standard sense, we could be dealing with a curved spacetime here.

-Black-hole: However, Mann and collaborators solved the equivalent of Einstein equations in their non-standard theory to find a particular form of the metric -a particular form of the function f in the notation of the authors of the current manuscript- and they are able to relate this particular form of the metric with black-hole physics, including its mass. However, the solution of Mann et al. is NOT the solution $f = \beta \tanh x$ considered by the authors. Therefore, even in this non-standard non-Einsteinian modification of gravity the metric proposed by the authors is not related to a black hole or gravity at all.

-Horizon: the authors certainly do show that there is a horizon. However, it is well-known that a horizon might appear in flat spacetime, for instance in the Rindler case, namely a uniformly accelerated observer through flat spacetime. Indeed, it is straightforward to see that the metric can be rewritten as $ds^2 = \beta^2 \tanh^2 x dt^2 - dx^2$ -by pulling out a conformal factor- and that $\tanh^2 x$ can be approximated by x^2 for small x . Then, we would have exactly the standard Rindler

metric, where beta would be the acceleration. This simple analysis suggests that the actual interpretation of the results should be more along the lines of acceleration, Unruh effect etc.

There is a very illustrative (wrong) answer of the authors to one point raised by Referee 3, who was asking about the mass of the simulated black hole. They say that:

"It might be difficult to define a mass for the two-dimensional black hole." This is wrong: mass would be easily defined if they have followed either a) or b) above.

Then they go on by -again, wrongly- using Hawking's formula to obtain the mass. This already assumes that the temperature they have computed is Hawking's temperature -which is not. Obviously, Hawking formula is derived within standard quantum field theory in curved spacetime with standard gravity and the Schwarzschild metric, and therefore is of no use here. The truth is that the authors do not know what is the black hole mass, simply because there is no mass, no black hole, no gravity or curved spacetime at all.

In summary, I have shown that the authors have not implemented an experiment to observe an analogue black-hole, curved spacetime or Hawking radiation. They have only generated a flat spacetime horizon, possibly related with Unruh physics. Therefore, I am afraid that this work cannot be published in a scientific journal in its current form.

Re: NCOMMS-22-15028-T

On-chip black hole: Hawking radiation and curved spacetime in a superconducting quantum circuit with tunable couplers

By Yun-Hao Shi, Run-Qiu Yang, Zhongcheng Xiang, *et al.*

In this letter, we provide the point-to-point responses and the corresponding revisions. We thank all Referees for the careful reading and useful comments on our work. We are enclosing the new version of our paper revised according to the comments and suggestions of the Referees. For convenience, the main changes are marked in blue in the additional manuscripts.

With best regards,
the authors.

Point-to-point responses and the corresponding revisions.

Report of Referee #1 -- NCOMMS-22-15028-T/Shi

Referee #1 commented:

I appreciate the authors' efforts to revise the manuscript in response of my and the other Referees' reports. Many technical points have been duly addressed and the manuscript has been purged by many (but not all, see below) flagrantly overselling statements.

Our response:

We thank the positive comments of Referee #1 on our efforts to revise the previous version of the manuscript. We will respond to additional comments in the report point by point as follows.

Referee #1 commented

Nonetheless, I am still not fully convinced about the scientific interest of a quantum simulation experiment where all complexities and the rich physics of the original problem are removed by complex mathematical manipulations (and quite strong hidden approximations) and one is left with a quite trivial single-particle propagation problem. In such an approach, what is interesting is the theory rather than the experiment. At odd with the authors' point of view, I still see a marked difference between the indirect nature of their observation of Hawking effect and the relatively direct evidence of Steinhauer's ones, with the consequent ease of addressing more subtle effects in this latter class of experiments.

Our response:

Firstly, Hawking radiation is a kinematic effect of quantum fields in curved spacetime with horizons and does not need to consider the self-interaction of quantum fields

[Schutz2009] and is independent of the gravity theory itself. Most of the interesting physics of Hawking radiation are completely described by free quantum fields in curved spacetime. One member of our team (Rong-Gen Cai) has been studying black hole physics for more than 30 years. For example, in Refs. [Cai2009, Cai2005], they studied Hawking radiation associated with the dynamical horizon of spacetime by using a quantum tunneling approach. These works are cited more than 250 and 630, respectively (the data are obtained from the website: <https://inspirehep.net/authors/1014768?ui-citation-summary=true>). We do believe our understanding of Hawking radiation is correct.

Secondly, though matters themselves have no self-interaction, the interaction between matter fields and curved spacetime exists, our mode is not a “trivial single-particle propagation” since the effects of curved spacetime have been encoded into the inhomogeneous hopping constant of lattices. Thus, this mapping does not remove the main physics of Hawking radiation or other effects of quantum field theory in curved spacetime.

In fact, the sprout of this idea can be traced back to at least 24 years ago, proposed by T. Jacobson [Jacobson1998, Jacobson2000], which is proposed even earlier than BEC model. T. Jacobson is a top expert in black hole physics, so his idea once attracted many people's attention. Lots of experimental progress have been made over the years in BEC approach, however, the experimental realization from the lattice approach has not made any progress before our work. From this fact, one can realize that the real challenge is how to realize a quantum lattice model experimentally so that couplings can be tuned based on the target curved spacetime and the system is directly measured. Two members of our team (Yang and Cai) with their collaborators first made an essential improvement to Jacobson's original idea [Jacobson1998, Jacobson1999] from a theoretical aspect (Ref. [18] in the main text). This work then presents the first experimental realization toward this ancient approach to the analogue of Hawking radiation. In a sense, we solved an experimental problem that has not progressed for more than 20 years.

In the revised manuscript, we emphasize the pioneering nature of Steinhauer's work in our main text: *“After years of development, the signature of analogue Hawking radiation based on density-density correlations has been reported in Bose-Einstein condensates”*. Here we just want to *“report an observation of analogue Hawking radiation on the superconducting quantum chip, which is also the first quantum realization of “lattice black hole” originally proposed by T. Jacobson more than twenty years ago”*. The corresponding model and experimental techniques are different from that of BEC. Our model is completely quantum. The evolution of entanglement is measurable in our model. Our experiment can reconstruct the density matrix, which contains information much more than density-density correlations. More importantly, we can design and control the parameters to match the various target curved spacetime precisely and so has the possibility of studying various effects of curved

spacetime in a unified platform, which is not easily realized in quantum fluid models.

In the response (see below, please) to summary part of Referee #1, we have listed a few interesting physical problems that are very suitable to be studied in our setup but very hard to be studied in Steinhauer's BEC or other quantum fluid models.

References:

- [Schutz2009] B. Schutz, A First Course in General Relativity, Cambridge University Press, Page 160, 2nd Edition (2009);
- [Cai2009] R. G. Cai, L. M. Cao and Y. P. Hu, Hawking Radiation of Apparent Horizon in a FRW Universe, *Class. Quant. Grav.* 26, 155018 (2009), [arXiv:0809.1554 [hep-th]].
- [Cai2005] R. G. Cai and S. P. Kim, First law of thermodynamics and Friedmann equations of Friedmann-Robertson-Walker universe, *JHEP* 02, 050 (2005), [arXiv:hep-th/0501055 [hep-th]].
- [Jacobson1998] S. Corley and T. Jacobson, Lattice black holes, *Phys. Rev. D* 57, 6269-6279 (1998) [arXiv:hep-th/9709166 [hep-th]].
- [Jacobson1999] T. Jacobson and D. Mattingly, Hawking radiation on a falling lattice, *Phys. Rev. D* 61, 024017 (1999) [arXiv:hep-th/9908099 [hep-th]].

Referee #1 commented

As I already mentioned in my first report, also the entanglement features studied here are mostly reduced via a Jordan-Wigner transformation to a free-fermion model, so I strongly doubt how such an experiment can capture much of the interesting entanglement features of a real BH. Unless the authors convince me that their platform can really help understanding the question raised by [34,37-39] or "the relationship between entropy and BH horizon area", I find the text in their conclusions as just empty marketing sentences.

Our response:

Firstly, Hawking radiation and its entanglement are indeed effects of free quantum field in curved spacetime [Hawking1975, Crispino2008]. The free theory can still present interesting entanglement features. One typical example in a lattice model is the "transverse-field Ising model" [Gu2010, Witten2018], which can be transformed into a free-fermion model in "momentum space" by using Jordan-Wigner transformation. However, such a model has a rich entanglement structure in real space. Particularly, the "quantum phase transition" can happen at the critical magnetic field, where a conformal field theory emerges.

Though questions such as the origin of black hole entropy and the information paradox raised by [34,37-39] cannot be answered completely by such a single analogy experiment, the black hole entropy is suspected to have a strong relationship to the entanglement of Hawking radiation and our analogy model has advantages in studying entanglement entropy. Particularly, density matrix tomography is a conventional operation in superconductivity [Kaixu2018, Yan2019]. On the contrary, other analogy

approaches are not easy to realize density matrix tomography and so not easy to study the evolution of entanglement. Thus, we think any progress in applying superconducting processors into analogy gravity will be meaningful to the above famous pending open problem.

To avoid misunderstanding and honestly present our work, we have removed the relative sentences involving Refs. [34,37-39] and "the relationship between entropy and BH horizon area". By the way, we like to mention that "the relationship between entropy and BH horizon area", namely, $S=A/4$, holds only in Einstein's general relativity. In another word, "the relationship between entropy and BH horizon area" involves the dynamics of gravity.

References:

[Hawking1975] S. W. Hawking, Particle creation by black holes, *Commun. Math. Phys.* **43**, 199–220 (1975);

[Crispino2008] Luís C. B. Crispino, Atsushi Higuchi, and George E. A. Matsas, The Unruh effect and its applications, *Rev. Mod. Phys.* **80**, 787-838 (2008);

[Witten2018] E. Witten, APS Medal for Exceptional Achievement in Research: Invited article on entanglement properties of quantum field theory, *Rev. Mod. Phys.* **90**, 045003 (2018); It also discusses many other lattice models which can be transformed into free fermion model but present rich entanglement structures in their ground states.

[Gu2010] Shi-Jian Gu, Fidelity approach to quantum phase transitions, *Int. J. Mod. Phys. B* **24**, 4371 (2010);

[Kaixu2018] Kai Xu, et. al, Emulating Many-Body Localization with a Superconducting Quantum Processor, *Phys. Rev. Lett.* **120**, 050507 (2018)

[Yan2019] Yan et al., Strongly correlated quantum walks with a 12-qubit superconducting processor, *Science* **364**, 753–756 (2019)

Referee #1 commented

To summarize, I am very hesitant about my recommendation for Nat.Comms. On one hand, I agree that the experiment is a clean and solid realization on a quantum simulator of a very simple model that may be connected via complicate maths to problems of actual research interest such as Hawking radiation. On the other hand, given how strongly the experiment relies on the preliminary mathematical manipulations, I can hardly see what further developments can arise from it in the direction of analog models.

Should the editor take a positive decision, I strongly recommend the authors to make a further willing effort to put the manuscript in a correct perspective with other analog models and down-tune the remaining overselling statements, as well to take the following comments into due consideration.

Our response:

We thank that the Referee agrees this work is a clean and solid realization on a

quantum simulator. As we explained, besides the Hawking radiation, many other effects also originate from free quantum fields in curved spacetime (note that many interesting efforts of quantum field theory in curved spacetime come from the interaction of quantum fields with curved spacetime, not from the self-interaction of quantum fields themselves). This is not because of the “complicated maths” used here. We have built the relationship between the spacetime geometry and lattice coupling, at the moment, we think at least there are following further developments that can arise from our setup in the direction of analog models:

- (1) One can excite two or more qubits state and observe how the spacetime curvature and horizon will influence the evolution of entanglement between (among) them;
- (2) The OTOC (Out-of-time-ordered correlators) is a very new important development in BH physics. Particularly, it has a strong relationship to the fast scrambling and bounds on the rate of information scrambling of BH. It is also interesting to study how the spacetime curvature and existence of the horizon will influence the evolution of OTOC. To measure the OTOC in analog models, one should simultaneously realize both forward $U(t)$ and backward $U(-t)$ continuous time evolution. This is easy to realize in our model by flipping the sign of Hamilton globally but is very hard to realize in quantum fluid models. Even in free lattice theory, the OTOC still has been shown interesting phenomenon. For example, Ref. [Green2022] simulated the OTOC for transverse-field Ising mode and found an interesting similarity to BH physics;
- (3) Though our model in this paper is designed for the (1+1)-dimensional case, in principle we can extend our theoretical model to explore the (1+2)-dimensional case (though in this case new theoretical efforts are needed and there is no boson-fermion correspondence). Since the spatial dimension becomes 2, we can easily design various different lattice networks of nontrivial topologies which appears in the study of BH physics;
- (4) We can consider the time-dependent background such as the famous Robertson-Walker metric, which is the most important metric to describe the dynamics of our universe.
- (5) In our setup, it is easy to consider a spacetime with two horizons such as Schwarzschild-de Sitter spacetime, which has not only the black hole horizon but also a cosmological horizon. To study the Hawking radiation of such spacetimes is certainly of great interest in our analog model.

The above aspects are interesting, not only for the study of gravity but also for condensed matter and quantum information, though they do not belong to the traditional contexts of analogue gravity.

As mentioned in Ref. [Maldacena2020]: “Simpler quantum systems display only some of the properties and more complex ones are believed to describe black holes in very concrete theories of Einstein gravity.” We believe that any effort is valuable in constructing such quantum systems both in theory and experiment.

However, we still completely agree with Referee that we should honestly assess our model only based on our current results. We follow the suggestions of the Referee to make more neutral comments about the Hawking radiation and simulation of the property of entanglement. We have tried our best to remove all the confused or overselling statements, and objectively conclude what this paper has achieved. For example, we deleted the sentences “*Our results would stimulate further interests to explore related problems such as information loss paradox of black holes*” and “*the relationship between entropy and black hole horizon area*” in the summary part.

References:

[Green2022] A. M. Green, A. Elben et al., Experimental Measurement of Out-of-Time-Ordered Correlators at Finite Temperature, Phys. Rev. Lett. **128**, 140601 (2022);
[Maldacena2020] J. Maldacena, Black holes and quantum information, Nature Reviews Physics **2**, 123 (2020);

Referee #1 commented

More technical remarks:

-Triggered by Referee 3's remark #6, I realized that the change in sign of κ plays no role at all, as it can be straightforwardly reverted to a positive κ (as in fig.11a) by a gauge-like transformation on the outside qu-bits, by alternatively rotating the spin state of the Q1,Q2 qubits by 0 or by π around z. This must be clear in the main text, otherwise readers may get misled.

Our response:

We thank Referee to emphasize this issue. We added the following sentence at the end of page 3:

“In fact, from the viewpoint of the lattice qubit model, the results will be equivalent if the function κ is replaced by $|\kappa|$ both in the case of curved and flat spacetime. Since we here map the coupling to the components of metric, the continuity requires κ changes the sign when passing through the analog horizon.”

Referee #1 commented

-Triggered again by Referee 3's remark #7, I would recommend the authors to include a discussion of the "continuum limit" which, in my opinion, is much more delicate than the finite-size effects of Sec.VII-C of the SM. In particular, both the tunneling effect and the localization on the horizon may be significantly modified when a finer grid of qubits is used to simulate the QFT.

Our response:

We thank Referee for raising this interesting issue. The “continuum limit” is the limit of keeping the physical length scale but decreasing lattice spacing to zero. The tunneling effect at the continuum limit has been shown in the theoretical paper of our one group (Ref. [18] of the main text). We added a Sec. VII D in the Supplementary Material to show localization on the horizon. Particularly, it shows that our model in

the classical, continuum limit and “geometrical optics approximation” can indeed reproduce the behaviors of outgoing lights inside the black holes.

Referee #1 commented

-The authors may comment the importance of the $(-i)^j$ factor in the definition of \tilde{c}_j between eq.(34) and (35) of the SM. This specific change of variables (together with the assumption of slowly varying c 's that is implicit in the continuum limit) is crucial to recover the correct form of the linear dispersion in the wave-equation (37). The authors should discuss these assumptions in full detail.

Our response:

The factor $(-i)^j$ is important to obtain the correct Heisenberg equation. This mathematical transformation is not first introduced by us. It has been widely used in the study of artificial lattices to simulate quantum fields in flat/curved spacetimes. We added three references and mentioned this point in the Supplementary Material.

Referee #1 commented

-It would be worth that the authors compare their model to the tunneling model developed in the analog model context in Phys. Rev. Lett. 94, 061302 (2005). This may be the closest atomic analog of their work.

Our response:

Thanks for your suggestion, we have added the following sentences near the end of page 5:

“The tunneling picture of Hawking radiation here is similar to the quantum fluid model of analog horizon [36] with two differences in details. The first one is that the analogue horizon of Ref. [36] is created by transonic flow but we here create analogue horizon by inhomogeneous lattice hopping. The second is that the injected beam of [36] is from the subsonic region (outside horizon) so that the reflected flow stands for the flow of Hawking radiation (classically the infalling beam should be swallowed by horizon completely and there is no reflected mode), but we here create a particle inside the horizon, so the transmission flow is the Hawking radiation.” (Ref. [36] in the revised main text is the paper Phys. Rev. Lett. 94, 061302 (2005))

In addition, the Letter [Phys. Rev. Lett. 94, 061302 (2005)] said “It is the aim of this Letter to show that indeed a 1D Fermi-degenerate noninteracting gas that scatters against a very smooth potential barrier provides a clear and straightforward quantum mechanical microscopic description of the sonic analog of Hawking radiation.” This also confirms what we have explained: Hawking radiation is the effect of non-interacting quantum fields in curved spacetime and the study on it does not invoke a self-interaction of the quantum field.

We thank Referee #1 again for all his/her comments, which are all valuable and very helpful for improving our paper, as well as the important guiding significance to our

research. We hope our above explanations and improvements could dispel the Referee's hesitation.

Report of Referee #3 -- NPHYS-2021-11-03172/Shi

Referee #3 commented:

The authors have reviewed their manuscript based on the reviewers' comments and resubmitted it to Nature Communications. I am very impressed with the level of detail shown by the authors in addressing my and the other reviewers' comments, and in my opinion, the manuscript is ready to be published in Nature Communications.

Our response:

In the previous reports, the Referee praised our experimental techniques in calibrating and controlling the coupler devices and provided practical suggestions that inspired us to improve our work. We appreciate the Referee very much for the recommendation.

Report of Referee #4 -- NPHYS-2021-11-03172/Shi

Referee #4 commented:

In this work, the authors claim to have built up experimentally an analogue of a black hole, including the detection of analogue Hawking radiation. They use an array of 10 superconducting transmon qubits which are coupled by 9 additional transmons operated as tunable couplers. The authors exploit an analogy between this system in the continuum limit -under strong dispersive conditions- and the dynamics of a Dirac field in a particular metric under particular coordinates - which the authors claim to be a curved-spacetime black-hole metric. The value of the couplings determine the existence of an horizon as a point where the coupling is 0 and have opposite signs at both sides of it. Analogue Hawking radiation would appear in the form of excited states of the qubit chain.

Unfortunately, there are several issues which prevent the interpretation of the results in the form claimed by the authors. Some of them -like the particular choice of coordinates needed- have already been discussed by previous referees. However, I believe that there are even more important issues.

Our response:

We first thank Referee #4 for his/her careful reading and detailed comments. Referee #4 thought there were "even more important issues". Let us now respond to them one by one.

Referee #4 commented:

- Curved spacetime: In standard Einstein gravity, it is a well-known fact that all 1+1 D spacetimes are conformally flat. Therefore, there are no 1+1 D curved spacetimes.

There are two alternatives to this fact: a) one can consider that the 1+1 D spacetime is just a section of a 3+1 D spacetime (see, for instance, NJP 20 053028 (2018)). This is not what the authors are doing, since their spatial coordinate is not a radial coordinate -it takes negative values inside the horizon. b) one can consider modifications of Einsteinian gravity, as in the papers of Mann and collaborators mentioned by the authors. Only in this non-standard sense, we could be dealing with a curved spacetime here.

Our response:

We thank Referee to raise these basic conceptions involved in general relativity and black hole physics. However, it seems that Referee's above descriptions are not accurate according to generally accepted understanding in the community of general relativity and black hole physics. For examples,

- (1) A conformally flat spacetime can still be curved spacetime. A spacetime is (locally) flat if and only if all components of Riemannian curvature tensor are zero, regardless it is conformal flat or not, and regardless of what gravitational theory is used (see Refs. [Schutz2009, Straumann2013] for examples). There are many curved 2-dimensional spacetimes. By the way, the 2-dimensional solution of Mann's theory is also locally conformally flat, since all 2-dimensional spacetimes are locally conformally flat, not only in standard Einstein gravity.
- (2) The modification of Einsteinian gravity of Mann is just one way to consider (1+1)-dimensional gravity theory rather than the "only way". There are also many different modifications, See Refs. [Jackiw1985, Teitelboim1983] for example. The two-dimensional JT gravity is under intensive study currently. Another well-known two-dimensional gravity model is the two-dimensional dilaton gravity from string theory. Note that the definition of black hole is determined by the causality of spacetime, and has nothing to do with the gravity theory itself. In addition, as mentioned above, Hawking radiation is a kinematical effect. Thus, once the metric is specified, one can study the Hawking radiation associated with the spacetime horizon and needs not to know in which gravity theory the metric is a solution.

In our case, after a few direct algebras, Referee can verify that the Riemannian curvature tensor in our model has nonzero components. Thus, our metric indeed describes a curved spacetime. We also know that physics is independent of the coordinates. Since x can only be finitely negative in our experimental chip, we can redefine the spatial coordinate by a trivial translation: $r = x + a$ with a positive constant a . Rewriting the metric in terms of r , the spatial coordinate then is non-negative. Thus, our model belongs to case *a*) mentioned by Referee.

We added the following sentence below Eq. (3) of main-text "*One can verify that this function $f(x)$ gives us nonzero Riemannian curvature tensor and so describes a 2-dimensional curved spacetime.*" to emphasize that we indeed study a curved 2-dimensional spacetime.

References:

[Schutz2009] B. Schutz, A First Course in General Relativity, Cambridge University Press, Page 160, 2nd Edition (2009);
[Straumann2013] N. Straumann, Affine Connections. In: General Relativity. Graduate Texts in Physics. (The section 15.9 theorem 16.5), Springer, Dordrecht (2013);
[Jackiw1985] R. Jackiw, Lower dimensional gravity, Nuclear Physics B **252**, 343-356 (1985);
[Teitelboim1983] C. Teitelboim, Gravitation and hamiltonian structure in two spacetime dimensions, Physics Letters B, **126**, 41-45 (1983).

Referee #4 commented:

-Black-hole: However, Mann and collaborators solved the equivalent of Einstein equations in their non-standard theory to find a particular form of the metric -a particular form of the function f in the notation of the authors of the current manuscript- and they are able to relate this particular form of the metric with black-hole physics, including its mass. However, the solution of Mann et al. is NOT the solution $f = \beta \tanh x$ considered by the authors. Therefore, even in this non-standard non-Einsteinian modification of gravity the metric proposed by the authors is not related to a black hole or gravity at all.

Our response:

The Referee could refer to Chapter 12.1 of Ref. [Wald1984] and the Chapter 5.4.1 of Ref. [Poisson2004] to find what are the definitions of the black hole and horizon. The referee then can see that the black hole is determined by only the causality of the metric itself but regardless of gravity theory. The metric in this paper indeed satisfies the definition of a horizon in curved spacetime.

Our paper only tried to simulate a few effects of the interesting properties of black hole rather than gravity theory itself. This artificial metric indeed captures the main physics of Hawking radiation as well as simplifies the experimental realization. Of course, other different metrics can also be realized in our platform. It is also very interesting to study various gravity theories in our platform in the future, including Mann's very particular gravity theory.

In addition, to explain why it is not necessary to follow the two ways mentioned by the Referee when we study 2-dimensional analogues black holes, let's give more examples. The first experimental measurement of analog Hawking radiation was reported by Ref. [Weinfurtner2011], which only created a sonic horizon; Refs. [Steinhauer2016, Kolobov2019] also created a 2-dimensional black hole. The Fig. 2 of Ref. [Steinhauer2016] and Fig. 1 of Ref. [Kolobov2019] show their models are very similar to ours: a horizon is at $x=0$ and there are two asymptotically flat regions in $x>0$ and $x<0$, respectively. None of Refs. [Weinfurtner2011, Steinhauer2016, Kolobov2019] justified that their models were a section of (3+1)-dimensional black hole or the solutions obtained by Mann's theory. In fact, such justification is not necessary since Hawking radiation is just the effects of the horizon and only the horizon is needed in the analog

of Hawking radiation. There is a complete but short review [Barceló2019] on the topic of analogue black-hole horizons which may be useful for the Referee. Various typical analogue black hole horizon models are reviewed in [Barceló2019] but none of them followed the requirements emphasized by Referee.

References:

- [Wald1984] R. M. Wald, General Relativity, The University of Chicago Press Chicago and London, Page 447 (1984);
- [Poisson2004] E. Poisson, A Relativist's Toolkit: The Mathematics of Black-Hole Mechanics, Cambridge University Press (2004);
- [Weinfurtner2011] S. Weinfurtner et al., Measurement of Stimulated Hawking Emission in an Analogue System, Phys. Rev. Lett. **106**, 021302 (2011);
- [Steinhauer2016] J. Steinhauer, Observation of quantum Hawking radiation and its entanglement in an analogue black hole. Nature Physics **12**, 959 (2016).
- [Kolobov2019] V. I. Kolobov et al., Observation of thermal Hawking radiation and its temperature in an analogue black hole. Nature **569**, 688-691 (2019);
- [Barceló2019] C. Barceló, Analogue black-hole horizons, Nature Physics **15**, 210-213 (2019).

Referee #4 commented:

-Horizon: the authors certainly do show that there is a horizon. However, it is well-known that a horizon might appear in flat spacetime, for instance in the Rindler case, namely a uniformly accelerated observer through flat spacetime. Indeed, it is straightforward to see that the metric can be rewritten as $ds^2 = \beta^2 \tanh^2 x dt^2 - dx^2$ -by pulling out a conformal factor- and that \tanh^2 can be approximated by x^2 for small x . Then, we would have exactly the standard Rindler metric, where β would be the acceleration. This simple analysis suggests that the actual interpretation of the results should be more along the lines of acceleration, Unruh effect etc.

Our response:

Firstly, we thank Referee that he found our metric is conformally flat. However, as we have clarified that a conformally flat spacetime can still be curved spacetime. The curvature in our model is nonzero, so our horizon is a horizon of curved spacetime rather than a horizon of flat spacetime.

Secondly, the metric $ds^2 = \beta^2 \tanh^2 x dt^2 - dx^2$ and $ds^2 = \beta \tanh x dt^2 - dx^2 / (\beta \tanh x)$ describe two completely different spacetimes since their curvature tensors are different though they are conformally equivalent to each other.

Thirdly, the near-horizon geometry of any non-extreme black hole always has a Rindler 2-dimensional sector (the pulling out a conformal factor is not necessary) and the black hole temperature indeed can be computed via Unruh temperature. The above discovery of the Referee is also one of the standard ways to derive the formula of Hawking temperature. See page 34 in the literature of "Lecture notes for a 'Part III'

course 'Black Holes' given in DAMTP, Cambridge" [Townsend1997]. However, black hole radiation and the Unruh effect are different in physics. Our model is not the Unruh effect since the spacetime is curved and there is a black hole.

References:

[Townsend1997] P. K. Townsend, Black Holes, arXiv:gr-qc/9707012v1;

Referee #4 commented:

There is a very illustrative (wrong) answer of the authors to one point raised by Referee 3, who was asking about the mass of the simulated black hole. They say that: "It might be difficult to define a mass for the two-dimensional black hole." This is wrong: mass would be easily defined if they have followed either a) or b) above. Then they go on by -again, wrongly- using Hawking's formula to obtain the mass. This already assumes that the temperature they have computed is Hawking's temperature -which is not. Obviously, Hawking formula is derived within standard quantum field theory in curved spacetime with standard gravity and the Schwarzschild metric, and therefore is of no use here. The truth is that the authors do not know what is the black hole mass, simply because there is no mass, no black hole, no gravity or curved spacetime at all.

Our response:

We first agree that the mass is well-defined if one considers the gravity theory following two approaches mentioned by Referee. However, the definition of the mass of black hole is a dynamic problem that depends on the gravity theory, but Hawking radiation is a kinematic problem that is determined by the metric itself. Different gravity theories have different ways to define mass. By the way, even the same metric could have different masses in different gravity theories. In our setup, since we did not specify the gravity theory, we could not give the exact mass associated with the metric.

Anyway, we still thank the Referee to raise this issue, since our original statement is a little misleading. We modified it as follows on page 5:

"It will depend on the gravity theories to define a mass for the two-dimensional black hole, for example, the Manns two-dimensional gravity theory [Mann1991] and Jackiw-Teitelboim gravity [Jackiw1985, Teitelboim1983] have different methods to define mass. However, since the analogue Hawking radiation is characterized by the temperature, we then give an estimation of how large a black hole in our real universe could reproduce the same temperature."

By the way, there are three very important works on the analogy of Hawking radiation, Refs. [Weinfurtner2011, Steinhauer2016, Kolobov2019], none of which followed two approaches required by Referee to build their analog models.

Here we would like to emphasize again that our model indeed describes a curved spacetime since the Riemannian curvature in our model is nonzero.

References:

- [Weinfurtner2011] S. Weinfurtner et al., Measurement of Stimulated Hawking Emission in an Analogue System, Phys. Rev. Lett. **106**, 021302 (2011);
- [Steinhauer2016] J. Steinhauer, Observation of quantum Hawking radiation and its entanglement in an analogue black hole. Nature Physics **12**, 959 (2016);
- [Kolobov2019] V. I. Kolobov et al., Observation of thermal Hawking radiation and its temperature in an analogue black hole. Nature **569**, 688-691 (2019);
- [Mann1991] R. B. Mann et al., Semiclassical gravity in 1+1 dimensions, Phys. Rev. D **43**, 3948 (1991);
- [Jackiw1985] R. Jackiw, Lower dimensional gravity, Nuclear Physics B **252**, 343-356 (1985);
- [Teitelboim1983] C. Teitelboim, Gravitation and hamiltonian structure in two spacetime dimensions, Physics Letters B, **126**, 41-45 (1983).

Referee #4 commented:

In summary, I have shown that the authors have not implemented an experiment to observe an analogue black-hole, curved spacetime or Hawking radiation. They have only generated a flat spacetime horizon, possibly related with Unruh physics. Therefore, I am afraid that this work cannot be published in a scientific journal in its current form.

Our response:

As we have clarified that the metric in our model has a nonzero Riemannian curvature tensor and so indeed describes curved spacetime. Hawking radiation is an effect of the horizon and does not depend on the gravity theory itself. All the analog models, such as Refs. [Weinfurtner2011, Steinhauer2016, Kolobov2019, Barceló2019], tried only to create an analog horizon, none of which follows the two ways a) and b) emphasized by Referee. The metric in our model is very similar to Refs. [Steinhauer2016, Kolobov2019]. We indeed generated a curved spacetime with a horizon and observed an analogue of Hawking radiation.

We hope all the above explanations and the following relevant references would be useful for the Referee to judge our manuscripts and to understand some relative conceptions in our work.

References:

- [Weinfurtner2011] S. Weinfurtner et al., Measurement of Stimulated Hawking Emission in an Analogue System, Phys. Rev. Lett. **106**, 021302 (2011);
- [Steinhauer2016] J. Steinhauer, Observation of quantum Hawking radiation and its entanglement in an analogue black hole. Nature Physics **12**, 959 (2016).
- [Kolobov2019] V. I. Kolobov et al., Observation of thermal Hawking radiation and its temperature in an analogue black hole. Nature **569**, 688-691 (2019);
- [Barceló2019] C. Barceló, Analogue black-hole horizons, Nature Physics **15**, 210-213 (2019).

Summary of changes

All our changes are marked in blue in the additional manuscripts. In the following we provide a summary of the changes:

Main text:

1. We added the sentence “In fact, from the viewpoint of lattice qubit model, the results will be equivalent if...” at the end of page 3;
2. We added a sentence below the Eq. (3) of main-text “One can verify that this function $f(x)$ gives us nonzero Riemannian curvature tensor and so describes a 2-dimensional curved spacetime.”;
3. We added “After years of development, the signature of analogue Hawking radiation based on density-density correlations...” at the end of page 4;
4. We have added a comparison with Ref. [Phys. Rev. Lett. 94, 061302 (2005)] mentioned by Referee #1 on page 5;
5. We have revised the expression as “To obtain the radiation probabilities, we...” on page 5;
6. We have added “It will depend on the gravity theories to define a mass for the two-dimensional black hole...”;
7. We have revised the conclusion part of our manuscript;
8. New references [22,30,31,37,38,39,40,44] have been added and cited; [37-39] in the original manuscripts were deleted.
9. We corrected typos/misprints.

Supplementary Material:

1. We have added “S” in front of all the numbers of equations and figures in Supplementary Material;
2. We have added “Note the factor $\$(-i)^j\$$ is important to obtain the correct Heisenberg equation. The similar trick is widely used in...”;
3. We have added the subsection D “Continuum limit” in Section VII;
4. Fig. S14 was added;
5. References [7-9] have been added and cited.

REVIEWER COMMENTS

Reviewer #4 (Remarks to the Author):

The authors have tried to refute the comments made in my first report, without introducing significant changes in their manuscript. Therefore, all my criticisms are still valid. I will try to be more clear this time:

- In 1+1 D and standard General Relativity, the Einstein tensor is always 0 and therefore all the solutions are vacuum solutions. Therefore, no 1+1 D gravity and obviously no black holes. I am sure the authors know this basic result, however I recall the pedagogical reference P. Collas Am. J. Phys. 45, 833 (1977). Of course, this fact is connected -while being obviously stronger- with the fact that all 1+1 metrics are conformally flat. The 1+1 curvature that the authors are referring to is a trivial one.

-The metric proposed by the authors is not a dimensional reduction of a 3+1 black-hole metric and is not a black-hole metric in Mann's theory. Is it a black hole metric in other gravity theory?(Btw. I never said that Mann's theory was "they only way" to do anything, as the authors misleadingly quote). The authors do not say. Therefore, the metric is not a black-hole metric.

- Despite the amount of irrelevant well-known references provided by the authors in their reply, in the experimental BEC black-hole analogues the equations of motion are related with the Schwarzschild metric in Gullstrand-Painlevé coordinates. In their manuscript they do not.

- The authors acknowledge that they are not simulating a black hole, but "a few effects of the interesting properties of black hole". More specifically, they are simulating a horizon and they think that this "indeed captures the main physics of Hawking radiation". They are wrong. The existence of an horizon and radiation does not entail the existence of Hawking radiation. For Hawking radiation, you need a black hole. Otherwise, the generation of radiation from the quantum vacuum could be related with different effects such as the Unruh effect or the Dynamical Casimir effect.

Summarizing, in this manuscript the authors propose a simulation of an horizon in a trivially curved spacetime within an analogue setup. However, they claim that they are simulating a black hole and Hawking radiation, which they are not. Therefore, this work cannot be published in a scientific journal.

The present manuscript describes an experimental study of analog gravity using a novel qubit-lattice setup based on superconducting transmon qubits. In particular, the authors observe the quantum walk and the scattering of an incident quasi-particle, claimed to be “analogue Hawking radiation” (see more technical observations below). They also make the experimental observation of the entanglement of the scattered state from an incident entangled state.

The experimental work behind the manuscript is exhaustive and detailed, and represents an impressive tour de force. The theoretical formalism behind the work is also quite remarkable, including thorough numerical simulations that strongly support the experimental data.

In general, the manuscript is expected to have a considerable impact since it is the first genuine lattice implementation of an analog gravity model. Given the rich tools and the high degree of control present in these setups along with their intrinsic quantum nature, the work can potentially inspire a whole line of research of analog gravity in qubit lattices, with promising perspectives.

Thus, based only on the above observations, the work would in principle deserve publication in *Nature Communications*. However, the manuscript contains a number of (unnecessary) overstatements and misinterpretations about the observed results that lack the scientific soundness expected from a high-impact article, which in addition inaccurately describe the physics behind their measurements. The work also presents some aspects that require further clarification. As a result, the manuscript is not suitable for publication, at least in its present form.

The main sources of inaccuracy are related to the claim of observation of analog Hawking radiation, and to the relevance and interpretation of the entanglement measurements. Indeed, already from the abstract, the work begins with an overstatement “However, due to the experimental difficulty of accurately constructing curved spacetime and precisely measuring the thermal spectrum, Hawking radiation and its quantum nature, such as entanglement, have not been adequately investigated”. This is not further developed later in the paper, so it is not clear why the current literature has not “adequately investigated” Hawking radiation, and where and how this work improves the current literature and in which sense (as will be seen from below, it is rather the opposite, at least regarding “quantum” Hawking radiation). We now explain in more detail these inaccuracies:

1) Regarding the claim of observation of analog Hawking radiation: In the field of analog gravity, it is essential to distinguish the nature of the Hawking process under study, whether it is *stimulated* or *spontaneous*, especially in experimental measurements. In the *stimulated* Hawking effect, an incoming (with respect to the horizon) channel is populated (for example, thermally or with some incident wave packet), and its scattering by the horizon *stimulates* the creation of positive-negative energy pairs, precisely those forming Hawking radiation. An important remark is that stimulated Hawking radiation can be typically described within a classical picture. In contrast, *spontaneous* Hawking radiation is the spontaneous emission of radiation by the event horizon in the complete absence of incoming waves, and it is a genuine quantum effect with no classical counterpart. Indeed, this is the original effect described by Hawking, and most of the times it is called just Hawking radiation (and sometimes *quantum* Hawking radiation to emphasize this aspect). In essence, Hawking radiation results from the fact

that the vacuum of the incoming modes is typically a squeezed state with respect to the outgoing modes, giving rise to spontaneous pair creation.

The above distinction is clearly done in most of the works in the literature. Indeed, the observations of [4,12] literally include the word “stimulated” in the very same title to leave clear the nature of their findings. However, the authors seem unaware of this critical issue, which leads to many misinterpretations of the experimental results. This is not only shown by the lack of a discussion “spontaneous vs. stimulated” within the manuscript, but also by the way in which the citations to the different experimental works are mixed in the introduction and everywhere else. The most critical point comes in the so-called “Observation of analogue Hawking radiation” section of the work. The authors state there: “After years of development, the signature of analogue Hawking radiation based on density-density correlations has been reported in Bose-Einstein condensates [10]. Here we report an observation of analogue Hawking radiation on the superconducting quantum chip...”

Now, [10] is the work by Steinhauer that did provide the first conclusive observation of *spontaneous* Hawking radiation (and the only one up to date, along with further measurements within the same BEC setup, as explained below), including the measurement of the Hawking temperature from the spectrum, in agreement with the theoretical and numerical predictions. Specifically, the spontaneous Hawking radiation was observed from the quantitative measurement of density-density correlations between the inside and the outside of the black hole, which describe the correlated emission of Hawking pairs from the horizon (the characteristic trait of Hawking radiation). The results of that work were further confirmed and expanded in the Nature Physics of 2021 also from Steinhauer, in which the spontaneous birth of Hawking correlations was experimentally observed, along with the predicted stationarity of the Hawking emission.

In the way it is written, the author’s claim that their work provides an “observation of analogue Hawking radiation” seems to be a claim about a *spontaneous* observation, a very strong statement that should be carefully justified in detail. Unfortunately, the authors do not only perform such a strong claim without justification, but also they later state: “This result can be considered as an important signature of Hawking radiation for the analog black hole [3, 4, 12, 32].” That is, the authors are invoking concepts and techniques from works based on stimulated Hawking radiation, such as [4,12], mixing once more the concepts of *stimulated* and *spontaneous* Hawking radiation without carrying such an important distinction.

This can seem just a question of semantics or presentation, but it is instead a fundamental discussion on the nature of the observed radiation, and a critical aspect of any analog experiment. Indeed, the natural question now is: of which kind is the experimental observation performed by the authors?

The answer, based on the above discussion, is clearly that the authors observe *stimulated* Hawking radiation, since in all experimental observations and numerical simulations they need to induce an excitation by flipping some qubits to 1 (either inside or outside the black hole) that eventually travel towards the horizon (see Figs. 2, 3). The authors do measure the Hawking temperature, but this does not mean that they are observing *spontaneous* Hawking radiation. Indeed, it is well known that the squared scattering coefficient $|\beta|^2$ governing the anomalous transmission of a *stimulated* process from inside to outside also determines the spontaneous spectrum of Hawking radiation, presenting a thermal-like dependence with the frequency. Thus, the Hawking temperature associated to a certain horizon can be extracted also

from the spectrum of stimulated processes without involving at any point *spontaneous* Hawking radiation. Actually, this aspect is not exclusive of analog gravity, and it also arises in other quantum systems such as atomic physics, where spontaneous emission is related to stimulated emission as already noted by Einstein.

If the observed Hawking radiation was indeed *spontaneous*, one would observe that, without inducing any excitation (Hawking radiation is a vacuum process), the horizon would be spontaneously emitting correlated pairs of quasi-particles, as was observed in [10] and Steinhauer's Nature Physics of 2021. This is not shown in any way by the experimental and numerical results from the authors.

The stimulated nature of the observed phenomenon is also shown by the entanglement measurements; see comment II) below. To observe entanglement, the authors have to change the experimental protocol with respect to the claimed observation of "analogue Hawking radiation". However, entanglement is intrinsic to the spontaneous Hawking effect, since this is originated by the squeezed nature of the vacuum (see the very recent PRA 104, 063302 (2021) and references therein for details). Hence, one would not have to modify the experiment giving rise to Hawking radiation for the entanglement observation if the former was truly spontaneous.

II) Regarding entanglement measurements: Quantum information aspects, such as entanglement, are critical in Hawking radiation, giving rise to the celebrated information paradox. That is why the study of the entanglement of (*spontaneous*) Hawking radiation is central in the field. Indeed, current lines of research are not only investigating the entanglement of Hawking radiation, but also the role of backreaction of Hawking radiation on the metric, essential for the understanding of black-hole evaporation.

However, none of these aspects can be studied in the setup proposed by the authors (at least, within the current model), making unfounded some of their statements. As explained above, the authors observe a particular form of stimulated Hawking radiation, which is not entangled. In fact, to observe entanglement in their analog setup, the authors need to start from an entangled state, whose scattering by the horizon is observed later. But the entanglement dynamics here is not spontaneously originated by the horizon or by any "gravitational" process: it is just present from the very beginning because the original state was entangled! Thus, the entanglement observed here has very little relation to the entanglement of Hawking radiation. Furthermore, no backreaction problem can be studied in this setup, as the effective background metric is independent of the quantum state of the system.

Now, the criticism raised in the previous paragraphs not only serves to correct a number of overstatements and misinterpretations by the authors but, on the bright side, it is also helpful to adequately inscribe the results of the work, which still are (as highlighted at the beginning of this review) impressive.

I) Regarding the claim of observation of analog Hawking radiation: Since now it is clear that the observed Hawking radiation is *stimulated*, the authors can correctly inscribe their work in the literature. Actually, while it is true that the observed radiation is stimulated Hawking radiation, this stimulation is genuinely quantum (indeed, it is based on a qubit), in contrast to the stimulated observations present in the literature in water waves and nonlinear light [4,12], which are fully classical. Therefore, this is a novel remarkable result from this work; see also comment II) below.

Moreover, I recommend to the authors to further emphasize that they are working with a fermionic analog, described by an effective (discrete) massless Dirac equation, in contrast to all present experiments, which work with bosonic analogs that simulate the massless Klein-Gordon equation for a scalar field. This is another interesting novel feature of the present work.

As a result of these comments, the authors should explicitly and correctly discuss the nature of their observation and the distinction between spontaneous and stimulated Hawking radiation, leaving quite clear the stimulated nature of their observation everywhere (including abstract, intro and conclusions), as it is done in all analog experiments, and how their findings are inscribed within the literature.

II) Regarding entanglement measurements: Even though the entanglement measurements presented here are not related with Hawking radiation and black-hole evaporation, they are quite interesting by themselves. First, they strongly support the claim above regarding the quantum character of the stimulated radiation, since they are able to create an incident entangled pair, something which is not possible in other analog setups within the current techniques (not even in condensates, where the observation of spontaneous Hawking radiation is in contrast possible). Actually, the concept of “entangled stimulated Hawking radiation” seems quite novel to me, and it is possible that it has not been even addressed in the vast theoretical literature on the topic (the authors could further investigate this). Also, this observation paves the way for future studies on entanglement, quantum information, or even many-body physics in curved spacetimes.

As a result of these comments, the authors should remove many incorrect statements which are unrelated to the nature of the observed entanglement (like the reference to the Page time in the Caption of Fig. 4, or the claim about the increase of the entanglement entropy “due to the Hawking radiation” in right column of Page 6) and correctly discuss the origin of their entanglement observation (unrelated to the spontaneous Hawking effect) and its significance.

Last but not least, I also have some comments on certain specific aspects of the manuscript which are not quite clear.

a) In general, the style and presentation of the manuscript can be improved, since there are many typos and even ill-constructed sentences. I recommend the authors to put some effort on improving the presentation of the manuscript.

b) Page 1, Introduction: “such as using supersonic fluid [2-7]”. The authors here mean “shallow water waves”, since essentially all analogs of black holes are based on some kind of subsonic-supersonic interface (or the equivalent optical concept).

c) Since \hbar is only set to one later, perhaps the authors should include \hbar in Eq. (1) or, even better, explain that they set $\hbar = c = 1$ before Eq. (1).

d) The role of d is quite confusing. On the one hand, they say that it is the lattice constant (after Eq. (2)) but, in the other hand, it later has “arbitrary units” (after Eq. (3)), and it enters in dimensionless functions like $f(x)$. Of course, after careful re-examination of all the calculations behind the formalism, one understands the role of d but, for a general reader, it can be quite confusing. In the way it is currently written, d can be only a dimensionless parameter, with no units, that controls the scale of variation of f over each lattice site. The authors should explain clearly this aspect in the manuscript.

e) Page 2, right column, after Eq. (2): “Here, the function $f(x)$...”

Is not the content of this sentence already explained in left column of the same page? The authors should remove this sentence to save space.

f) At the end of both Page 2 and Page 3, the authors include a quite obscure discussion on the role of κ that contain many confusing statements and even ill-constructed sentences. I attribute this to a presentation problem more than to a conceptual problem; see comment a) above. My recommendation here is that the authors devote some effort to rewrite those sentences to explain clearly what they really mean in both places.

g) Page 4, left column: What is quenched in the experimental setup? The on-site potential μ , the coupling κ or both? The on-site potential is set in all places to the same frequency ω_{ref} ? Moreover, what is exactly the “ground state of a qubit”? The ground state is a global concept of a system, not a local one. Perhaps they mean that 0 is annihilated by σ_- ? The present redaction is quite confusing. Therefore, the authors should explain clearly the details of the transient and of the formation of the analog setup in this paragraph.

h) Page 5, right column: I have seen that, due to the comment of one of the Referees, the authors devote a significant portion of the Letter to justify the use of a 1+1 D model. In my view, this is not needed, since most of the canonical experiments on analog gravity are indeed implemented in a 1+1 D model, e.g. those by Steinhauer, and it is well-established that they can be used to study a number of analog phenomena, including Hawking radiation.

I agree with the Referee that the analogy should not be overextended beyond its applicability (for example, as explained, any reference to the Page time should be erased as there is no evaporation) but, for instance, the correspondence that the authors make between their measured Hawking temperature and the mass of an astrophysical object is indeed adequate, since it helps to give a qualitative picture to the general reader that is expected to be the target of *Nature Communications*. Indeed, this kind of comparison has been already carried out in the mentioned Nature Physics of 2021 by Steinhauer. Precisely, an interesting point would be to compare the measured Hawking temperature in this work with those present in the literature, since the former is several orders of magnitude higher. I leave this to the authors criterion.

As a result, the authors can save space in the main text by removing (some of) the sentences used to justify the validity of their metric.

i) The role of the “pulse sequences” labeled by $k=1,2,\dots$ in the measurements is not clear at all in the manuscript, see captions of Figs. 2, 4. They are easily confused with the pulse sequences giving rise to the entangled pair (Fig. 1c). The physical meaning and the specific role played by the pulse sequences $k=1,2,\dots$ in the measurements should be explained in the main text.

j) In page 6, left column: It is not fully clear which quantum state is used for the computation of the entanglement entropy and the concurrence. I guess that in both cases it is the two-qubit quantum state ρ_{in} formed by qubits Q_1, Q_2 but the statement about entanglement entropy “quantifying the entanglement between the interior of black hole and the exterior” is again confusing (notice that there is no entanglement due to Hawking radiation here and no evaporation is taking place). Therefore, the authors should rewrite this paragraph to explain clearly the magnitudes used to study entanglement and avoid confusing statements.

k) In the same place, the technical definition of the concurrence is not really needed as it is a well-known concept and can be safely cited from the literature, saving space in this way.

l) Supp. Mat., Page 1, left column: should not be $\{t,x\}$ instead of $\{v,x\}$?

m) SM, Section III: Due to the pedagogical nature of the presentation of the model (another added value of the work), for generality, I would recommend to the authors that they use a general index N to label the number of sites instead of fixing it to their particular value $N=10$.

n) The discussion around Eq. (S39) is poorly written (in general, the authors should also try to improve the presentation of the SM).

o) Final suggestion: An interesting result of the SM is that the authors can construct an analog setup with two horizons. Now, configurations with two horizons contain a lot of rich physics and are a current subject of research in the analog field. In particular, while the observation of spontaneous Hawking radiation of [10] is now accepted by the community, the black-hole laser observation published by Steinhauer in 2014 was disputed and later explained by Jacobson and co-workers in terms of experimental fluctuations of some background Cherenkov wave, something supported and extended by Steinhauer's observations in the Nature Physics of 2021. Therefore, a future perspective of this work could be to study stimulated radiation between two horizons, which could potential shed some light on the black-hole laser problem, an effect not yet conclusively observed.

In summary, the authors should remove all the overstatements and misinterpretations of their results (which are indeed unnecessary given the quality of the work) and correctly discuss how the main findings inscribe within the literature, following the discussions on spontaneous vs. stimulated Hawking radiation and entanglement (labeled as I) and II), respectively) provided above. Finally, they should also address the comments a)-o) above.

Re: NCOMMS-22-15028A

On-chip black hole: Hawking radiation and curved spacetime in a superconducting quantum circuit with tunable couplers

By Yun-Hao Shi, Run-Qiu Yang, Zhongcheng Xiang, *et al.*

In this letter, we provide point-to-point responses and corresponding revisions. We thank all Reviewers for the careful reading and useful comments on our work. We are enclosing the new version of our paper revised according to the comments and suggestions of the Reviewers. The main changes are marked in blue in the additional manuscript for convenience.

With best regards,
the authors.

Point-to-point responses and the corresponding revisions.

Report of Reviewer #4 -- NCOMMS-22-15028A/Shi

Reviewer #4 commented:

The authors have tried to refute the comments made in my first report, without introducing significant changes in their manuscript. Therefore, all my criticisms are still valid. I will try to be more clear this time:

Our response:

We thank the Reviewer for his/her time and patience. In the following, let us respond to the Reviewer's comments point-to-point.

Reviewer #4 commented:

- In 1+1 D and standard General Relativity, the Einstein tensor is always 0 and therefore all the solutions are vacuum solutions. Therefore, no 1+1 D gravity and obviously no black holes. I am sure the authors know this basic result, however I recall the pedagogical reference P. Collas Am. J. Phys. 45, 833 (1977). Of course, this fact is connected -while being obviously stronger- with the fact that all 1+1 metrics are conformally flat. The 1+1 curvature that the authors are referring to is a trivial one.

Our response:

Firstly, we thank the Reviewer offers us a useful reference, and we agree with two facts mentioned by the Reviewer: *i*) 1+1 D spacetime has zero Einstein tensor and *ii*) is (locally) conformally flat (in all gravity theories). We are sorry that we cannot explain in detail the definition of black holes on this occasion, but we still strongly recommend the following two references that we mentioned in the last reply:

- (1) P.K. Townsend, Black Holes, arXiv:gr-qc/9707012v1;
- (2) Robert M. Wald (1984), "General Relativity", The University of Chicago Press Chicago and London

which explain what a black hole is clearly from mathematical and physical aspects. A black hole is defined by its spacetime causal structure. Roughly speaking, a black hole is a region covered by an event horizon. Please pay attention on this fact that when one considers analogue black holes, usually, one needs only specify the metric of the black hole spacetime, and does not need know the gravity theory itself where the black hole spacetime is a solution of the gravity theory. On the other hand, after we carefully read the reference [P. Collas Am. J. Phys. 45, 833 (1977)] we can only learn the following conclusion: *in Einstein's theory, 1+1 D spacetime can have curvature but not matter*. This is of course true. Therefore, in 1+1 D spacetime, the Einstein's theory does not make any sense. This is also one of facts that in 1+1 D gravity theory, one has to introduce other fields, like JT gravity discussed below and dilaton gravity theory extensively discussed in the literature. As a fact, one can easily check that our metric indeed describes a nontrivial black hole spacetime as we discussed in our manuscript. We do hope our simple explanation above clarifies the referee's confusion.

Reviewer #4 commented:

-The metric proposed by the authors is not a dimensional reduction of a 3+1 black-hole metric and is not a black-hole metric in Mann's theory. Is it a black hole metric in other gravity theory?(Btw. I never said that Mann's theory was "they only way" to do anything, as the authors misleadingly quote). The authors do not say. Therefore, the metric is not a black-hole metric.

Our response:

We partially agree with the referee: our metric is not a black hole metric in Mann's theory. But our metric could be a dimensional reduction of a 3+1 dimensional black hole metric. Clearly by adding an angular part, our metric could be uplifted to 3+1 dimensions. As we argued above, in our manuscript, we need not specify the gravity theory since we are not going to study the dynamics of gravity itself, while we are interested in the dynamics of a free scalar field (fermion field) in the metric. Here let us say a little more about the 1+1 D gravity theory. For a 1+1-dimensional metric $ds^2 = -f(r)dt^2 + dr^2/f(r)$, we can regard it to be a solution of JT gravity [Roman Jackiw, *Lower dimensional gravity*, Nuclear Physics B, Volume 252, 1985, Pages 343-356; Claudio Teitelboim, *Gravitation and hamiltonian structure in two spacetime dimensions*, Physics Letters B, Volume 126, Issues 1-2, 1983, Pages 41-45]. JT gravity is not an Einstein theory and can contain matter. One can also find the nonzero energy-momentum tensor from our metric for JT gravity. Except for JT gravity, there are many different 2D non-Einstein gravity theories including the dilaton gravity mentioned above. It is straightforward to find suitable matter distributions so that our metric becomes the solution for such gravity theories, so we did not explain this issue in the manuscript. More importantly, as we have explained in the previous response, the black hole is completely defined by the metric itself (causal structure) and it is completely irrelevant to discuss which gravity theory can give such a metric. Our metric describes a black hole. To confirm this statement, one does not need to specify any gravity theory. Instead, one can directly check our metric by analyzing the causal structure of the spacetime.

Reviewer #4 commented:

- Despite the amount of irrelevant well-known references provided by the authors in their reply, in the experimental BEC black-hole analogues the equations of motion are related with the Schwarzschild metric in Gullstrand-Painlevé coordinates. In their manuscript they do not.

Our response:

Though we think the issues raised by the Reviewer in the previous report are interesting, we offered many well-known references in the previous reply to prove that what the Reviewer required is not necessary for the analogy of Hawking radiation. The Reviewer said “in the experimental BEC black-hole analogues the equations of motion are related with the Schwarzschild metric in Gullstrand-Painlevé coordinates”, however, to the best of our knowledge, this is not completely true. Let us make an analysis here.

The Schwarzschild metric in Gullstrand-Painlevé coordinates for the 2-dimensional case takes the following form

$$ds^2 = -\left(1 - \frac{2M}{r}\right) dt^2 + 2\sqrt{\frac{2M}{r}} dt dr + dr^2.$$

The results in [Barcelo, Carlos et al. Living Rev.Rel.8:12,2005, gr-qc/0505065] show that the effective metric of fluid reads

$$ds^2 = -(c_s^2 - c^2) dt^2 + 2c dt dr + dr^2.$$

Here c_s is the acoustic velocity, and c is the speed of the fluid. The metric of the BEC is related to “Schwarzschild metric in Gullstrand-Painlevé coordinates” only if the speed is given by

$$c \propto \frac{1}{\sqrt{r}}.$$

For example, in the paper [Steinhauer, J. Observation of quantum Hawking radiation and its entanglement in an analogue black hole. *Nature Phys* 12, 959–965 (2016)], Fig. 2 explains the speed distribution in the BEC. It does not satisfy such requirements. In the paper [Muñoz de Nova, J.R., Golubkov, K., Kolobov, V.I. et al. Observation of thermal Hawking radiation and its temperature in an analogue black hole. *Nature* 569, 688–691 (2019)], Fig. 2b shows the speed of the fluid. If the Reviewer reads this paper, he/she could clearly see from Fig. 2b that the speed of the fluid can be described approximately by the formula $c = c_1 \tanh x/x_0 + c_0$ with three positive constants c_0, c_1 and x_0 , which is very similar to what we did but does not satisfy the requirement of “Schwarzschild metric in Gullstrand-Painlevé coordinates”. The authors in these two papers did not use any sentence to discuss if their 2-dimension metric is any solution to gravity theories. In addition, in the first experimental work of analogue Hawking radiation [Weinfurter, Silke et al., *Measurement of Stimulated Hawking Emission in an Analogue System*, *Phys. Rev. Lett.* 106, 021302], the effective metric is created by experimental apparatus shown in its Fig. 2. If the Reviewer reads this paper, he/she will find that the speed of water depends on the shape of the obstacle and even cannot be described by a simple mathematical formula. It also does not obey the requirement of “Schwarzschild metric in Gullstrand-Painlevé coordinates”. Finally let us mention the fact that the key to realize the Hawking radiation in BEC experiment is there exists a black hole horizon where $c_s = c$,

in one side of the horizon, one has $c_s < c$, while in another side of the horizon, one has $c_s > c$.

In the above three relevant well-known references, all of them only created an analogue horizon of two-dimensional spacetime. None of them even uses any piece sentence to mention if the metric is a solution of some gravity theory or reduced from a higher dimensional black hole. The detailed mathematical expression of metric is even irrelevant since they only require that there is a horizon with finite surface gravity.

Reviewer #4 commented:

- The authors acknowledge that they are not simulating a black hole, but "a few effects of the interesting properties of black hole". More specifically, they are simulating a horizon and they think that this "indeed captures the main physics of Hawking radiation". They are wrong. The existence of an horizon and radiation does not entail the existence of Hawking radiation. For Hawking radiation, you need a black hole. Otherwise, the generation of radiation from the quantum vacuum could be related with different effects such as the Unruh effect or the Dynamical Casimir effect.

Our response:

Firstly, to the best of our knowledge, there have not been any analogue experiments that can simulate all properties of black holes. Our sentence "a few effects of the interesting properties of black hole" just honestly presents what we did, and this is also what all other analogue experiments can claim at most. Here our description may cause some confusion, what we are saying is simulating the Hawking radiation of the analogue black hole. Secondly, it is true that the generation of radiation from the quantum vacuum could be related to different effects, such as the Unruh effect or the dynamical Casimir effect. However, the Unruh effect appears for an accelerating observer in flat spacetime and the dynamical Casimir effect appears in spacetime with time-dependent boundary conditions. Our system simulates a black hole metric, which is curved and time-independent spacetime, so the radiation cannot be the Unruh effect or the dynamic Casimir effect. Thirdly, as we have explained above, our 1+1D metric indeed describes a black hole.

Reviewer #4 commented:

Summarizing, in this manuscript the authors propose a simulation of an horizon in a trivially curved spacetime within an analogue setup. However, they claim that they are simulating a black hole and Hawking radiation, which they are not. Therefore, this work cannot be published in a scientific journal.

Our response:

We thank the Reviewer again for his/her time and patience. All in all, our work indeed simulated an analogue black hole in a lattice model. The site-dependent coupling distribution reflects the curved spacetime background with nonzero Riemannian curvature, which is indeed a black hole spacetime. The tunneling of the quasi-particles from the interior of the horizon to the outside corresponds to the analogue Hawking radiation, where the Hawking temperature is measured via the blackbody spectrum. We have revised our paper extensively for a clearer presentation. We hope all the explanations in

the current and previous response letters would be useful for the Reviewer to understand our setup and the significance of the analogue gravity experiments.

Report of Reviewer #5 -- NCOMMS-22-15028A/Shi

Reviewer #5 commented:

The present manuscript describes an experimental study of analog gravity using a novel qubit-lattice setup based on superconducting transmon qubits. In particular, the authors observe the quantum walk and the scattering of an incident quasi-particle, claimed to be “analogue Hawking radiation” (see more technical observations below). They also make the experimental observation of the entanglement of the scattered state from an incident entangled state.

The experimental work behind the manuscript is exhaustive and detailed, and represents an impressive tour de force. The theoretical formalism behind the work is also quite remarkable, including thorough numerical simulations that strongly support the experimental data.

In general, the manuscript is expected to have a considerable impact since it is the first genuine lattice implementation of an analog gravity model. Given the rich tools and the high degree of control present in these setups along with their intrinsic quantum nature, the work can potentially inspire a whole line of research of analog gravity in qubit lattices, with promising perspectives.

Thus, based only on the above observations, the work would in principle deserve publication in Nature Communications. However, the manuscript contains a number of (unnecessary) overstatements and misinterpretations about the observed results that lack the scientific soundness expected from a high-impact article, which in addition inaccurately describe the physics behind their measurements. The work also presents some aspects that require further clarification. As a result, the manuscript is not suitable for publication, at least in its present form.

Our response:

We thank the Reviewer for his/her praise of our work. We have extensively revised the manuscript according to the professional and enlightening comments of the Reviewer, as seen in the following responses. All our changes are marked in blue in the manuscripts attached.

Reviewer #5 commented:

The main sources of inaccuracy are related to the claim of observation of analog Hawking radiation, and to the relevance and interpretation of the entanglement measurements. Indeed, already from the abstract, the work begins with an overstatement “However, due

to the experimental difficulty of accurately constructing curved spacetime and precisely measuring the thermal spectrum, Hawking radiation and its quantum nature, such as entanglement, have not been adequately investigated". This is not further developed later in the paper, so it is not clear why the current literature has not "adequately investigated" Hawking radiation, and where and how this work improves the current literature and in which sense (as will be seen from below, it is rather the opposite, at least regarding "quantum" Hawking radiation). We now explain in more detail these inaccuracies:

Our response:

We thank the Reviewer for this comment. We have deleted the sentence "However, due to..." in the abstract.

Reviewer #5 commented:

1) Regarding the claim of observation of analog Hawking radiation: In the field of analog gravity, it is essential to distinguish the nature of the Hawking process under study, whether it is stimulated or spontaneous, especially in experimental measurements. In the stimulated

Hawking effect, an incoming (with respect to the horizon) channel is populated (for example,

thermally or with some incident wave packet), and its scattering by the horizon stimulates the creation of positive-negative energy pairs, precisely those forming Hawking radiation.

An important remark is that stimulated Hawking radiation can be typically described within a classical picture. In contrast, spontaneous Hawking radiation is the spontaneous emission of radiation by the event horizon in the complete absence of incoming waves, and it is a genuine quantum effect with no classical counterpart. Indeed, this is the original effect described by Hawking, and most of the times it is called just Hawking radiation (and sometimes quantum Hawking radiation to emphasize this aspect). In essence, Hawking radiation results from the fact that the vacuum of the incoming modes is typically a squeezed state with respect to the outgoing modes, giving rise to spontaneous pair creation.

The above distinction is clearly done in most of the works in the literature. Indeed, the observations of [4,12] literally include the word "stimulated" in the very same title to leave clear the nature of their findings. However, the authors seem unaware of this critical issue, which leads to many misinterpretations of the experimental results. This is not only shown by the lack of a discussion "spontaneous vs. stimulated" within the manuscript, but also by the way in which the citations to the different experimental works are mixed in the introduction and everywhere else. The most critical point comes in the so-called "Observation of analogue Hawking radiation" section of the work. The authors state there: "After years of development, the signature of analogue Hawking radiation based on density-density correlations has been reported in Bose- Einstein condensates [10]. Here we report an observation of analogue Hawking radiation on the superconducting quantum chip..."

Now, [10] is the work by Steinhauer that did provide the first conclusive observation of spontaneous Hawking radiation (and the only one up to date, along with further measurements within the same BEC setup, as explained below), including the measurement of the Hawking temperature from the spectrum, in agreement with the theoretical and numerical predictions. Specifically, the spontaneous Hawking radiation was observed from the quantitative measurement of density-density correlations between the inside and the outside of the black hole, which describe the correlated emission of Hawking pairs from the horizon (the characteristic trait of Hawking radiation). The results of that work were further confirmed and expanded in the *Nature Physics* of 2021 also from Steinhauer, in which the spontaneous birth of Hawking correlations was experimentally observed, along with the predicted stationarity of the Hawking emission.

In the way it is written, the author's claim that their work provides an "observation of analogue Hawking radiation" seems to be a claim about a spontaneous observation, a very strong statement that should be carefully justified in detail. Unfortunately, the authors do not only perform such a strong claim without justification, but also they later state: "This result can be considered as an important signature of Hawking radiation for the analog black hole [3, 4, 12, 32]." That is, the authors are invoking concepts and techniques from works based on stimulated Hawking radiation, such as [4,12], mixing once more the concepts of stimulated and spontaneous Hawking radiation without carrying such an important distinction.

This can seem just a question of semantics or presentation, but it is instead a fundamental discussion on the nature of the observed radiation, and a critical aspect of any analog experiment. Indeed, the natural question now is: of which kind is the experimental observation performed by the authors?

The answer, based on the above discussion, is clearly that the authors observe stimulated Hawking radiation, since in all experimental observations and numerical simulations they need to induce an excitation by flipping some qubits to 1 (either inside or outside the black hole) that eventually travel towards the horizon (see Figs. 2, 3). The authors do measure the Hawking temperature, but this does not mean that they are observing spontaneous Hawking radiation. Indeed, it is well known that the squared scattering coefficient $|\beta|^2$ governing the anomalous transmission of a stimulated process from inside to outside also determines the spontaneous spectrum of Hawking radiation, presenting a thermal-like dependence with the frequency. Thus, the Hawking temperature associated to a certain horizon can be extracted also from the spectrum of stimulated processes without involving at any point spontaneous Hawking radiation. Actually, this aspect is not exclusive of analog gravity, and it also arises in other quantum systems such as atomic physics, where spontaneous emission is related to stimulated emission as already noted by Einstein.

If the observed Hawking radiation was indeed spontaneous, one would observe that, without inducing any excitation (Hawking radiation is a vacuum process), the horizon

would be spontaneously emitting correlated pairs of quasi-particles, as was observed in [10] and Steinhauer's *Nature Physics* of 2021. This is not shown in any way by the experimental and numerical results from the authors.

The stimulated nature of the observed phenomenon is also shown by the entanglement measurements; see comment II) below. To observe entanglement, the authors have to change the experimental protocol with respect to the claimed observation of "analogue Hawking radiation". However, entanglement is intrinsic to the spontaneous Hawking effect, since this is originated by the squeezed nature of the vacuum (see the very recent PRA 104, 063302 (2021) and references therein for details). Hence, one would not have to modify the experiment giving rise to Hawking radiation for the entanglement observation if the former was truly spontaneous.

II) Regarding entanglement measurements: Quantum information aspects, such as entanglement, are critical in Hawking radiation, giving rise to the celebrated information paradox. That is why the study of the entanglement of (spontaneous) Hawking radiation is central in the field. Indeed, current lines of research are not only investigating the entanglement of Hawking radiation, but also the role of backreaction of Hawking radiation on the metric, essential for the understanding of black-hole evaporation. However, none of these aspects can be studied in the setup proposed by the authors (at least, within the current model), making unfounded some of their statements. As explained above, the authors observe a particular form of stimulated Hawking radiation, which is not entangled. In fact, to observe entanglement in their analog setup, the authors need to start from an entangled state, whose scattering by the horizon is observed latter. But the entanglement dynamics here is not spontaneously originated by the horizon or by any "gravitational" process: it is just present from the very beginning because the original state was entangled! Thus, the entanglement observed here has very little relation to the entanglement of Hawking radiation. Furthermore, no backreaction problem can be studied in this setup, as the effective background metric is independent of the quantum state of the system. Now, the criticism raised in the previous paragraphs not only serves to correct a number of overstatements and misinterpretations by the authors but, on the bright side, it is also helpful to adequately inscribe the results of the work, which still are (as highlighted at the beginning of this review) impressive.

Our response:

We thank the Reviewer for the professional and enlightening comments. Particularly, we thank the Reviewer to raise this crucial issue that was not well clarified in the manuscript. We indeed learned much from the comments of the Reviewer. We have modified and adjusted some comments of references, including Refs. [4,12] of our previous version. After we repeatedly checked our experiment and considered the comments of the Reviewer, we found crucial details in our experimental realization. In our experiment, the couplings are not determined by the structure of the quantum chip but by real-time control according to the program. In the section "Quantum walks in analogue curved spacetime" of our previous version, we wrote,

“Once the initial state is prepared, we apply the rectangle Z pulses on each qubit to make the on-site potential μ_j of all qubits at a reference frequency $\omega_{\text{ref}}/(2\pi) \approx 5.1$ GHz. During the quench dynamics, the hopping couplings κ_j between qubits are fixed as Eq.(3) (curved spacetime) or a constant (flat spacetime) via biasing couplers.”

Thus, in our experiment, we first excite the qubit and then tune couplings into the desired values. In other words, we prepare the particle before the horizon appears. As shown in Fig.1c, the evolution in the curved (or flat) spacetime starts after preparing the initial state.

After the horizon forms, the system evolves spontaneously without any extra excitations.

The physical picture is as follows: after the horizon appears, the system spontaneously evolves; the energy inside the horizon decreases but energy outside increases; the observers outside the horizon will find spontaneous energy flux coming from the black hole. The so-called stimulated emission of black hole is very different from this picture. It is the emission of an existing black hole caused by excitation outside the black hole. For example, in Ref. [4], sub-supersonic regions have been ready first, and then the excitations are generated. Based on these truths and after carefully considering the issues raised by the Reviewer, we think this subtle but important detail shows that it is seemingly not suitable to regard our result as “stimulated Hawking radiation”, but it is a spontaneous radiation.

Let us talk about more why we should first prepare particles (energy) before the black hole forms. Hawking radiation is an effect that the energy of a black hole emits into infinity spontaneously. We see that there are two aspects: there is a black hole, and the black hole should contain energy. For a real black hole, the interior of the horizon always contains energy since the black hole is usually formed by dense matters. For the BEC setup, though the analogue black hole is not formed by dense matters, the fluid itself always contains various available energies. Thus, in the BEC setup, we only need to create a horizon. However, in our chip model, the black hole geometry and energy inside the black hole need to be created independently. Tuning couplings to a particular distribution just creates a black hole geometry, but the whole system is still devoid of energy if we do not excite any qubit. In our experiment, we first prepare a particle then we tune the coupling to form the horizon that wraps up this particle. This is the process of black hole formation in our model but not the process of Hawking radiation. Once our black hole geometry is ready, the system evolves spontaneously. The particle inside the black hole *spontaneously* tunnels into the outside. This “tunneling picture” is equivalent to the picture of “pair creation” from the vacuum. In a word: *we induce an excitation by flipping some qubits to $|1\rangle$ before the black hole appears, but the radiation is spontaneous after the black hole appears.* We have highlighted this in the revised manuscripts. We thank the Reviewer for inspiring us to think more deeply about our experiments. Given our experimental design described above, it seems to have just avoided suspicions of the stimulated Hawking radiation. The observed Hawking radiation here is indeed spontaneous.

To study the entanglement, we changed the initial state, since we indeed want to study how the horizon affects the evolution of the entangled Bell pair. This is just our purpose,

so we prepare the Bell state $(|00\rangle + |11\rangle)/\sqrt{2}$ as the initial state. During the quench dynamics, component $|00\rangle$ does not evolve, and thus the evolution of entanglement entropy is dependent on the evolution of $|11\rangle$ governed by the Hamiltonian Eq. (2). Here, $|11\rangle$ still tends to radiate out through the Hawking radiation mechanism. Therefore, we use Hawking radiation to explain the non-trivial growth of the entanglement entropy in this evolution.

In addition, we added the sentence “The Hawking radiation of a black hole is spontaneous in nature. The first realization of spontaneous Hawking radiation in an analogue experiment was in BEC system [8]” at the beginning of the section “Observation of analogue Hawking radiation” to emphasize the spontaneous nature of Hawking radiation. We thank the Reviewer again for bringing up this crucial fact.

Reviewer #5 commented:

I) Regarding the claim of observation of analog Hawking radiation: Since now it is clear that the observed Hawking radiation is stimulated, the authors can correctly inscribe their work in the literature. Actually, while it is true that the observed radiation is stimulated Hawking radiation, this stimulation is genuinely quantum (indeed, it is based on a qubit), in contrast to the stimulated observations present in the literature in water waves and nonlinear light [4, 12], which are fully classical. Therefore, this is a novel remarkable result from this work; see also comment II) below. Moreover, I recommend to the authors to further emphasize that they are working with a fermionic analog, described by an effective (discrete) massless Dirac equation, in contrast to all present experiments, which work with bosonic analogs that simulate the massless Klein-Gordon equation for a scalar field. This is another interesting novel feature of the present work. As a result of these comments, the authors should explicitly and correctly discuss the nature of their observation and the distinction between spontaneous and stimulated Hawking radiation, leaving quite clear the stimulated nature of their observation everywhere (including abstract, intro and conclusions), as it is done in all analog experiments, and how their findings are inscribed within the literature.

Our response:

As we explained above, the observed Hawking radiation here is spontaneous. We did not make essential modifications to our statement concerning the main conclusions and explanations. However, the professional comments regarding “stimulated or spontaneous” Hawking radiation is thought-provoking, which are necessary and illustrative for the results of our work.

We thank the Reviewer for the suggestion about emphasizing the fermionic analogue. We have added the sentence “we report a fermionic lattice-model-type realization of an analogue black hole” in the abstract.

In an extended sense, we think there are three different perspectives to help understand whether the radiation here is spontaneous or stimulated:

(1) From the perspective of gravity theory, the typical stimulated radiation is the black hole

superradiance or the stimulated Hawking radiation discussed by reference [4]. There are two characteristics in the stimulated radiation: the radiation is caused by exterior (outside horizon) excitations, which are imposed after the black hole horizon has formed. However, our model excites the qubits before the horizon appears and spontaneously evolves after the black hole is ready. The radiation process is spontaneous though the formation of black hole is not.

- (2) From the perspective of quantum optics, the stimulated radiation is induced by radiation fields. Mathematically, the operator of a radiation field can be expressed as $g(b + b^\dagger)$ (single-mode) or $\sum_k g_k(b_k + b_k^\dagger)$ (multi-mode). However, after the preparation of the initial state, the total Hamiltonian of our system only contains the time-independent (but site-dependent) interaction between qubits without the radiation field term. Thus, in our setup, we think the radiation here is spontaneous.
- (3) From the perspective of laser principle, the laser system is a typical stimulated radiation system. An important feature in laser systems is the negative temperature (non-equilibrium) state due to the continuous population inversion. However, our system has no continuous population inversion, and the observed Hawking radiation here shows a thermal equilibrium spectrum.

Reviewer #5 commented:

II) Regarding entanglement measurements: Even though the entanglement measurements presented here are not related with Hawking radiation and black-hole evaporation, they are quite interesting by themselves. First, they strongly support the claim above regarding the quantum character of the stimulated radiation, since they are able to create an incident entangled pair, something which is not possible in other analog setups within the current techniques (not even in condensates, where the observation of spontaneous Hawking radiation is in contrast possible). Actually, the concept of “entangled stimulated Hawking radiation” seems quite novel to me, and it is possible that it has not been even addressed in the vast theoretical literature on the topic (the authors could further investigate this). Also, this observation paves the way for future studies on entanglement, quantum information, or even many-body physics in curved spacetimes. As a result of these comments, the authors should remove many incorrect statements which are unrelated to the nature of the observed entanglement (like the reference to the Page time in the Caption of Fig. 4, or the claim about the increase of the entanglement entropy “due to the Hawking radiation” in right column of Page 6) and correctly discuss the origin of their entanglement observation (unrelated to the spontaneous Hawking effect) and its significance.

Our response:

As we explained above, the evolution of entanglement entropy depends on the evolution of component $|11\rangle$ of the Bell state. It is suitable to use Hawking radiation to explain the non-trivial growth of the entanglement entropy in this evolution. To show more clearly the spontaneous nature of Hawking radiation in our work, we modified Fig.1c:

and the corresponding caption:

“Experimental pulse sequence for observing dynamics of entanglement, which consists of three parts, i.e., (I) initialization, (II) evolution, and (III) measurement. For the initialization (I), we prepare an entangled Bell pair on Q_1 and Q_2 by combining several single-qubit pulses and a two-qubit control-phase (CZ) gate. At the left boundary of region (II), the curved (or flat) spacetime forms. Then the system will evolve according to the corresponding κ_j in the Hamiltonian for a time t . In region (III), we perform the QST measurement.”

The pulse sequences of the quantum walk and observing Hawking radiation are similar to this, but the initial state is different or QST measurement is to be done.

In addition, we have deleted the reference to the Page time in the Caption of Fig. 4 according to the suggestion of the Reviewer.

Reviewer #5 commented:

Last but not least, I also have some comments on certain specific aspects of the manuscript which are not quite clear.

a) In general, the style and presentation of the manuscript can be improved, since there are many typos and even ill-constructed sentences. I recommend the authors to put some effort on improving the presentation of the manuscript.

b) Page 1, Introduction: “such as using supersonic fluid [2-7]”. The authors here mean “shallow water waves”, since essentially all analogs of black holes are based on some kind of subsonic-supersonic interface (or the equivalent optical concept).

c) Since \hbar is only set to one later, perhaps the authors should include \hbar in Eq. (1) or, even better, explain that they set $\hbar = c = 1$ before Eq. (1).

Our response:

We thank the Reviewer for the careful reading. We have revised the manuscript according to these suggestions. All our changes are marked in blue in the extensively revised manuscripts attached.

Reviewer #5 commented:

d) The role of d is quite confusing. On the one hand, they say that it is the lattice constant (after Eq. (2)) but, in the other hand, it later has “arbitrary units” (after Eq. (3)), and it enters in dimensionless functions like $f(x)$. Of course, after careful re-examination of all the calculations behind the formalism, one understands the role of d but, for a general reader, it can be quite confusing. In the way it is currently written, d can be only a dimensionless parameter, with no units, that controls the scale of variation of f over each lattice site. The authors should explain clearly this aspect in the manuscript.

Our response:

The role of d is embodied in discrete spatial position $x = x_j = (j - j_h)d$, where j is the index of the qubit. Here, d is the lattice constant, which has a dimension of distance. In the original Eq. (3), we made d dimensionless in order to match the dimensions of the coupling strength. To avoid ambiguity, we modified Eq. (3) in the revised manuscript:

$$\kappa_j = \frac{\beta \tanh((j - j_h)\eta d)}{4\eta d},$$

where η controls the scale of variation of f over each lattice site, which has the dimension of $1/d$. Here, we fix $\eta d = 0.35$ a.u. in the analogue curved spacetime experiment. We thank the Reviewer for pointing out this issue. We have explained this in both the main text and supplementary.

Reviewer #5 commented:

*e) Page 2, right column, after Eq. (2): “Here, the function $f(x)$...”
Is not the content of this sentence already explained in left column of the same page? The authors should remove this sentence to save space.*

Our response:

Thanks for this suggestion. To save space, we have moved the detailed description of the metric (Page 2, left column) to the Methods and reserved the necessary statement of the metric here.

Reviewer #5 commented:

f) At the end of both Page 2 and Page 3, the authors include a quite obscure discussion on the role of κ that contain many confusing statements and even ill-constructed sentences. I attribute this to a presentation problem more than to a conceptual problem; see comment a) above. My recommendation here is that the authors devote some effort to rewrite those sentences to explain clearly what they really mean in both places.

Our response:

We thank the Reviewer for the careful reading. We noticed that some descriptions did not distinguish well between the experimental system and the physical problem. We have carefully revised the paper to address this issue, which should not confuse readers now.

Reviewer #5 commented:

g) Page 4, left column: What is quenched in the experimental setup? The on-site potential μ , the coupling κ or both? The on-site potential is set in all places to the same frequency ω_{ref} ? Moreover, what is exactly the “ground state of a qubit”? The ground state is a global concept of a system, not a local one. Perhaps they mean that 0 is annihilated by σ^- ? The present redaction is quite confusing. Therefore, the authors should explain clearly the details of the transient and of the formation of the analog setup in this paragraph.

Our response:

All qubits are initially at their idle frequencies, where the frequency differences (detunings) are much larger than couplings, so they are almost decoupled from each other. Here we denote $|0\rangle$ and $|1\rangle$ as the “ground state” and “excited state” of a qubit at idle frequency. For a more direct description, we modified on Page 4 that “...with $|0\rangle$ and $|1\rangle$ being the eigenstates of $\hat{\sigma}_j^+ \hat{\sigma}_j^-$ ”. After preparing the initial state, both the on-site potential and coupling are quenched in the experimental setup. We have improved the presentation here: “We apply the rectangular Z pulses on all qubits to quench them in resonance at a reference frequency of $\omega_{\text{ref}}/(2\pi) \approx 5.1$ GHz. Meanwhile, the hopping coupling κ_j is fixed as Eq. (3) (curved spacetime) or a constant (flat spacetime) by controlling couplers”.

Reviewer #5 commented:

h) Page 5, right column: I have seen that, due to the comment of one of the Referees, the authors devote a significant portion of the Letter to justify the use of a 1+1 D model. In my view, this is not needed, since most of the canonical experiments on analog gravity are indeed implemented in a 1+1 D model, e.g. those by Steinhauer, and it is well-established that they can be used to study a number of analog phenomena, including Hawking radiation.

Our response:

We thank the Reviewer for this suggestion. We have removed the relevant content from the revised manuscript.

Reviewer #5 commented:

I agree with the Referee that the analogy should not be overextended beyond its applicability (for example, as explained, any reference to the Page time should be erased as there is no evaporation) but, for instance, the correspondence that the authors make between their measured Hawking temperature and the mass of an astrophysical object is indeed adequate, since it helps to give a qualitative picture to the general reader that is expected to be the target of Nature Communications. Indeed, this kind of comparison has been already carried out in the mentioned Nature Physics of 2021 by Steinhauer. Precisely, an interesting point would be to compare the measured Hawking temperature in this work with those present in the literature, since the former is several orders of magnitude higher. I leave this to the authors criterion.

As a result, the authors can save space in the main text by removing (some of) the sentences used to justify the validity of their metric.

Our response:

We thank the Reviewer for this suggestion. We have added the following paragraph to the revised main text to compare measured Hawking temperature and equivalent mass of the analogue black hole with that from other experiments:

“If we consider a Schwarzschild black hole in four-dimensional spacetime with the same Hawking temperature T_H , its mass can be calculated by $M/M_s \approx 6.4 \times 10^{-8} K/T_H$ [Hawking1974], where $M_s \approx 2 \times 10^{30}$ kg is the solar mass. For the simulated black hole in our work, $M/M_s \sim 10^{-3}$, whereas the typical value reported in BEC system for this quantity can be $\sim 10^2$. The significant difference in magnitude is attributed to the range of interaction energy manipulated in different experimental systems. In superconducting qubits, the coupling strength is usually on the order of MHz and thus the analogue surface gravity g_h is of the same magnitude, leading to $T_H = g_h/(2\pi) \sim 10^{-5}$ K. Differently, the interaction energy in BEC systems is $\sim 10^1$ Hz, corresponding to $T_H \sim 10^{-10}$ K [Steinhauer2010, Steinhauer2019, Isoard2020, Steinhauer2021]. In a much lower energy system like shallow water wave, T_H is even only about 10^{-12} K [Weinfurtner2011].”

In addition, we tabulated these for comparison in Supplementary Information, see Table S2.

As for the description of the metric, we have moved it to the Methods for saving space.

References:

- [Hawking1974] S. W. Hawking, Black hole explosions? *Nature* **248**, 30 (1974);
[Steinhauer2010] O. Lahav et al., Realization of a Sonic Black Hole Analog in a Bose-Einstein Condensate, *Phys. Rev. Lett.* **105**, 240401 (2010);
[Steinhauer2019] V. I. Kolobov et al., Observation of thermal Hawking radiation and its temperature in an analogue black hole, *Nature* **569**, 688-691 (2019);
[Isoard2020] M. Isoard et al., Departing from Thermality of Analogue Hawking Radiation in a Bose-Einstein Condensate, *Phys. Rev. Lett.* **124**, 060401 (2020);
[Steinhauer2021] V. I. Kolobov et al., Observation of stationary spontaneous Hawking radiation and the time evolution of an analogue black hole, *Nature Physics* **17**, 362 (2021);
[Weinfurtner2011] S. Weinfurtner et al., Measurement of Stimulated Hawking Emission in an Analogue System, *Phys. Rev. Lett.* **106**, 021302 (2011);

Reviewer #5 commented:

i) The role of the “pulse sequences” labeled by $k=1,2\dots$ in the measurements is not clear at all in the manuscript, see captions of Figs. 2, 4. They are easily confused with the pulse sequences giving rise to the entangled pair (Fig. 1c). The physical meaning and the specific role played by the pulse sequences $k=1,2\dots$ in the measurements should be explained in the main text.

Our response:

We thank the Reviewer for pointing out this problem. Here “ $k=1, 2\dots$ pulse sequences” means “repetitive experimental runs”. The error bars of the data, indicating the standard deviations, are calculated from the measurement results of these repetitive experimental runs. To avoid confusion, we have changed it in the revised manuscript.

Reviewer #5 commented:

j) In page 6, left column: It is not fully clear which quantum state is used for the computation of the entanglement entropy and the concurrence. I guess that in both cases it is the two-qubit quantum state ρ_{in} formed by qubits Q1,Q2 but the statement about entanglement entropy “quantifying the entanglement between the interior of black hole and the exterior” is again confusing (notice that there is no entanglement due to Hawking radiation here and no evaporation is taking place). Therefore, the authors should rewrite this paragraph to explain clearly the magnitudes used to study entanglement and avoid confusing statements.

k) In the same place, the technical definition of the concurrence is not really needed as it is a well-known concept and can be safely cited from the literature, saving space in this way.

Our response:

These two quantities are indeed calculated from the two-qubit state ρ_{in} of Q₁ and Q₂. The reason why the entanglement entropy obtained in this way quantifies the entanglement inside and outside the black hole is as follows:

The state of the total system (the interior of black hole and the exterior) is always a pure state during the evolution. Thus, the entanglement entropy of the two subsystems: $S(\rho_{in}) = S(\rho_{out})$, which quantifies the entanglement contained in this bipartite quantum system [Vidal2003, Amico2008].

Usually, the entanglement between the interior and exterior of a black hole is quantified by $S(\rho_{out})$ [Fiola1994, Yang2020]. However, in our experiment, the cost of measuring ρ_{out} is higher than measuring ρ_{in} due to the dimension of the Hilbert space. Therefore, we take advantage of the equivalence between $S(\rho_{in})$ and $S(\rho_{out})$, and thus compute $S(\rho_{in})$ to measure the entanglement between the interior and exterior of a black hole.

For a clearer presentation, we have applied the Reviewer’s suggestion and rewritten this paragraph. More technical definitions were moved into the Methods.

References:

[Vidal2003] G. Vidal et al., Entanglement in Quantum Critical Phenomena, Phys. Rev. Lett. **90**, 227902 (2003);

[Amico2008] L. Amico et al., Entanglement in many-body systems, Rev. Mod. Phys. **80**, 517 (2008);

[Fiola1994] G. Vidal et al., Black hole thermodynamics and information loss in two dimensions, Phys. Rev. D **50**, 3987 (1994);

[Yang2020] R.-Q. Yang et al., Simulating quantum field theory in curved spacetime with quantum many-body systems, Phys. Rev. Research **2**, 023107 (2020);

Reviewer #5 commented:

l) *Supp. Mat., Page 1, left column: should not be $\{t,x\}$ instead of $\{v,x\}$?*

m) *SM, Section III: Due to the pedagogical nature of the presentation of the model (another added value of the work), for generality, I would recommend to the authors that they use a general index N to label the number of sites instead of fixing it to their particular value $N=10$.*

n) *The discussion around Eq. (S39) is poorly written (in general, the authors should also try to improve the presentation of the SM).*

Our response:

We thank the Reviewer for the careful reading and detailed revision suggestions. We have made changes in these places and improved the presentation of Supplementary Information.

Reviewer #5 commented:

o) *Final suggestion: An interesting result of the SM is that the authors can construct an analog setup with two horizons. Now, configurations with two horizons contain a lot of rich physics and are a current subject of research in the analog field. In particular, while the observation of spontaneous Hawking radiation of [10] is now accepted by the community, the black-hole laser observation published by Steinhauer in 2014 was disputed and later explained by Jacobson and co-workers in terms of experimental fluctuations of some background Cherenkov wave, something supported and extended by Steinhauer's observations in the Nature Physics of 2021. Therefore, a future perspective of this work could be to study stimulated radiation between two horizons, which could potential shed some light on the black-hole laser problem, an effect not yet conclusively observed.*

Our response:

We would like to thank the Reviewer for his/her interest and prospect for the future development of our work. In fact, using our superconducting platform with tunable couplers, we can engineer arbitrary coupling distributions in a single quantum chip. In our future work, we expect to explore more aspects of black hole physics, such as the black hole laser problem mentioned by the Reviewer.

Summary of changes

All our changes are marked in blue in the additional manuscripts. In the following we provide a summary of the main changes:

Main text:

1. The Methods section has been added, including “Metric of two-dimensional spacetime”, “Tunable effective couplings”, and “Measurement of entanglement”;
2. The titles of some sections are slightly changed: “Quantum walks in analogue curved spacetime”, and “Dynamics of an entangled pair in the analogue black hole”;
3. We modified Fig. 1c and changed its caption;
4. We deleted “However, due to the experimental difficulty...” in Abstract and reorganized the rest;
5. We replaced “supersonic fluid” with “shallow water waves” in the Introduction;
6. The detailed description of the metric was moved to the Methods;
7. We have modified Eq. (3);
8. We have added the sentence “The Hawking radiation of a black hole is spontaneous in nature. The first realization of spontaneous Hawking radiation in an analogue experiment was in BEC system.”;
9. The generation of the Bell state and definitions of entanglement entropy and concurrence were moved to the Methods;
10. New references [10,11] have been added and cited; [39,40,42] in the original manuscripts were deleted.
11. We corrected typos/misprints.

Supplementary Information:

1. The word “Supplementary Material” has been changed to “Supplementary Information”;
2. “ $\{v,x\}$ ” in Section I was replaced by “ $\{t,x\}$ ”;
3. We have used a general index N to label the number of sites instead of fixing it to their particular value $N=10$ in Section III;
4. We have improved the discussion around Eq. (S39);
5. We have modified Eq. (S43);
6. Subsection “VII. E. Comparison of measured Hawking temperature with other experiments” has been added.
7. Table S2 has been added.

The authors have considerably improved the presentation of the manuscript and addressed many of the issues raised in previous reports. However, before publication, there are still some inaccuracies and overstatements that need to be fixed.

My main comment concerns the strong claim about the “spontaneous” nature of the Hawking radiation (HR) observed, which is completely inaccurate, as I will explain below.

I begin by noticing that, regardless of the picture used to describe Hawking radiation (tunneling, scattering, pair creation, Unruh effect...), what is spontaneous and what is stimulated is quite clear. If the process arises from the pure vacuum, where no radiation is present, it is spontaneous since it happens in the complete absence of excitations. If the creation is induced by the incidence on the horizon of radiation (such as typically coherent or thermal radiation), then it is stimulated. In any actual Hawking process, there will be some stimulated contribution which, nevertheless, can be distinguished from the spontaneous one if the former is not sufficiently strong, as done in the recent experiments by Steinhauer.

In this work, the authors prepare an initial state by setting one qubit inside the black hole to $|1\rangle$, and then create the horizon. This is stated in Page 4, right column: “Here we emphasize that the horizon appears after the initial state is ready. There are no additional excitations in the evolution” (by the way, I would like to notice that the early observation of spontaneous HR of Ref. [8] is not conventionally regarded as conclusive, as compared to later works of Steinhauer).

However, later in Page 5, left column, they claim: “It is also need to emphasize that we induce an excitation by flipping a qubit to $|1\rangle$ before the black hole appears, but the radiation is spontaneous after the black hole appears”, which is the main source of inaccuracy.

As inferred by the above considerations, the Hawking radiation observed here is clearly stimulated, since the authors need to stimulate the emission by inducing some excitation in the form of an excited qubit $|1\rangle$. For instance, in a condensate, the process would be equivalent to excite a phonon travelling towards the horizon. The fact that the horizon is formed after inducing the excitation is irrelevant for this matter.

Spontaneous HR would correspond to the case where the horizon spontaneously emits radiation without the need of any stimulation (that is, when the initial state is the ground state with all qubits set to $|0\rangle$). However, as the authors state in their response to the reviewers, this is not possible within this type of setup since “in our chip model, the black hole geometry and energy inside the black hole need to be created independently. Tuning couplings to a particular distribution just creates a black hole geometry, but the whole system is still devoid of energy if we do not excite any qubit”.

Thus, this kind of setup cannot exhibit the spontaneous Hawking effect, in contrast to other quantum systems, where one does not need to induce any excitation to observe HR.

Therefore, the authors should remove the statements “Here we emphasize that the horizon appears after the initial state is ready. There are no additional excitations in the evolution.” and “It is also need to emphasize that we induce an excitation by flipping a qubit to $|1\rangle$ before the black hole appears, but the radiation is spontaneous after the black hole appears”, and related ones, and place a proper discussion on the nature of the Hawking radiation observed. Nevertheless, as I suggested in my previous report, this is not necessarily a weakness of this work since, precisely, the stimulated HR observed here is of a novel kind as it is genuinely quantum. The equivalent in a condensate or an optical system could be for example to induce a

pure number state and observe the stimulated HR resulting from its scattering on the horizon, something quite challenging and indeed, a quite challenging experiment. Thus, the fact that one can work with stimulated HR of quantum nature is quite novel and should be emphasized by the authors, something which makes even less necessary the presence of any misleading overstatement on the spontaneous nature of the observation.

On the other hand, the methods and concepts behind the work are far clearer now, and I appreciate the effort done by the authors. As a result of the better understanding of the manuscript, I have some comments:

1) The authors now introduce a parameter η to make the function $f(x)$ truly dimensionless with $\eta d = 0.35$ a dimensionless parameter that controls the variation of f through the lattice. However, precisely because ηd is dimensionless (η has explicitly units of $1/d$, as the authors state), it has no units, so they should remove the arbitrary units label "a.u." in both the main text and the Supplementary Information.

2) Page 5, left column: "Assuming that $|E_n\rangle$ is the n -th eigenenergy of total Hamiltonian and ρ_{out} is the density matrix outside obtained by QST, then the probability of finding a particle of energy E_n outside the horizon can be obtained as $P_n = \langle E_n | \rho_{out} | E_n \rangle$ ". If $|E_n\rangle$ lives in the 10-qubit dimensional Hilbert space and ρ_{out} is the reduced density matrix outside the horizon, living in a 7-qubit Hilbert space, how exactly is evaluated P_n ? Because the dimensions of the bracket do not agree between the density matrix and the states. This should be further clarified, if not in the text, at least in the Methods section.

3) Page 5, right column: While referring to the experiments with shallow water waves in a tank, the authors state "In a much lower energy system like shallow water wave". Well, I would not say that a water tank at room temperature is a low energy system as compared to a condensate close to zero temperature... The authors should rewrite properly this sentence. Perhaps they mean in a "fully macroscopic setup" (since the Hawking temperature is determined by the scales of the setup; that is why they have such a low T_H)?

4) In Page 6, left column, it is not very clear why do they mean by "the entanglement between the pair in the black hole is protected by the analogue gravity so that it can slow the decrease of the concurrence". This is a very misleading statement. Most likely, the authors mean that the Hawking radiation stimulated at the horizon prevents the loss of entanglement?

5) The authors work with two density matrices ρ_{in}, ρ_{out} to describe the interior and exterior of the black hole. However, they miss one qubit, the one at the horizon. Then, it is no longer true $S(\rho_{out}) = S(\rho_{in})$, even if the initial state is pure. While this is not critical, since $S(\rho'_{out}) = S(\rho_{in})$, with ρ'_{out} also containing Q_3 , the authors should correct the statement about $S(\rho_{out}) = S(\rho_{in})$ and leave more clear this aspect.

6) Last but not least, in general, I would further encourage the authors to polish the text of the manuscript.

Re: NCOMMS-22-15028B

On-chip black hole: Hawking radiation and curved spacetime in a superconducting quantum circuit with tunable couplers

By Yun-Hao Shi, Run-Qiu Yang, Zhongcheng Xiang, *et al.*

In this letter, we provide the point-to-point responses and the corresponding revisions. We thank Reviewer #5 for his/her careful reading and useful comments on our work. We are enclosing the new version of our paper revised according to the comments and suggestions of Reviewer #5. For convenience, the main changes are marked in blue in the additional manuscripts.

With best regards,
the authors.

Point-to-point responses and the corresponding revisions.

Report of Reviewer #5 -- NCOMMS-22-15028B/Shi

Reviewer #5 commented:

The authors have considerably improved the presentation of the manuscript and addressed many of the issues raised in previous reports. However, before publication, there are still some inaccuracies and overstatements that need to be fixed.

Our response:

We thank the Reviewer for his/her suggestions for improving our manuscript. We have revised the manuscript according to the suggestions, as seen in the following responses. All our changes are marked in blue in the manuscripts attached.

Reviewer #5 commented:

My main comment concerns the strong claim about the “spontaneous” nature of the Hawking radiation (HR) observed, which is completely inaccurate, as I will explain below.

I begin by noticing that, regardless of the picture used to describe Hawking radiation (tunneling, scattering, pair creation, Unruh effect...), what is spontaneous and what is stimulated is quite clear. If the process arises from the pure vacuum, where no radiation is present, it is spontaneous since it happens in the complete absence of excitations. If the creation is induced by the incidence on the horizon of radiation (such as typically coherent or thermal radiation), then it is stimulated. In any actual Hawking process, there will be some stimulated contribution which, nevertheless, can be distinguished from the spontaneous one if the former is not sufficiently strong, as done in the recent experiments by Steinhauer.

In this work, the authors prepare an initial state by setting one qubit inside the black hole to 1, and then create the horizon. This is stated in Page 4, right column: “Here we emphasize that the horizon appears after the initial state is ready. There are no additional excitations in the evolution” (by the way, I would like to notice that the early observation of spontaneous HR of Ref. [8] is not conventionally regarded as conclusive, as compared to later works of Steinhauer).

However, later in Page 5, left column, they claim: “It is also need to emphasize that we induce an excitation by flipping a qubit to $|1\rangle$ before the black hole appears, but the radiation is spontaneous after the black hole appears”, which is the main source of inaccuracy.

As inferred by the above considerations, the Hawking radiation observed here is clearly stimulated, since the authors need to stimulate the emission by inducing some excitation in the form of an excited qubit $|1\rangle$. For instance, in a condensate, the process would be equivalent to excite a phonon travelling towards the horizon. The fact that the horizon is formed after inducing the excitation is irrelevant for this matter.

Spontaneous HR would correspond to the case where the horizon spontaneously emits radiation without the need of any stimulation (that is, when the initial state is the ground state with all qubits set to $|0\rangle$). However, as the authors state in their response to the reviewers, this is not possible within this type of setup since “in our chip model, the black hole geometry and energy inside the black hole need to be created independently. Tuning couplings to a particular distribution just creates a black hole geometry, but the whole system is still devoid of energy if we do not excite any qubit”.

Thus, this kind of setup cannot exhibit the spontaneous Hawking effect, in contrast to other quantum systems, where one does not need to induce any excitation to observe HR.

Therefore, the authors should remove the statements “Here we emphasize that the horizon appears after the initial state is ready. There are no additional excitations in the evolution.” and “It is also need to emphasize that we induce an excitation by flipping a qubit to $|1\rangle$ before the black hole appears, but the radiation is spontaneous after the black hole appears”, and related ones, and place a proper discussion on the nature of the Hawking radiation observed. Nevertheless, as I suggested in my previous report, this is not necessarily a weakness of this work since, precisely, the stimulated HR observed here is of a novel kind as it is genuinely quantum. The equivalent in a condensate or an optical system could be for example to pure number state and observe the stimulated HR resulting from its scattering on the horizon, something quite challenging and indeed, a quite challenging experiment. Thus, the fact that one can work with stimulated HR of quantum nature is quite novel and should be emphasized by the authors, something which makes even less necessary the presence of any misleading overstatement on the spontaneous nature of the observation.

Our response:

We thank the Reviewer for the professional comments. We understand that the excitation

by flipping a qubit in the preparation of the initial state is undeniable. As the reviewer pointed out, the process in our experiment does not arise from the pure vacuum.

We have deleted the sentence “Here we emphasize that the horizon appears after the initial state is ready. There are no additional excitations in the evolution.” and “It is also need to emphasize that we induce an excitation by flipping a qubit to $|1\rangle$ before the black hole appears, but the radiation is spontaneous after the black hole appears”.

Moreover, in the abstract and the corresponding section of the revised manuscript, we have clarified that the nature of the analogue Hawking radiation observed here is “stimulated”.

Reviewer #5 commented:

On the other hand, the methods and concepts behind the work are far clearer now, and I appreciate the effort done by the authors. As a result of the better understanding of the manuscript, I have some comments:

1) The authors now introduce a parameter η to make the function $f(x)$ truly dimensionless with $\eta d = 0.35$ a dimensionless parameter that controls the variation of f through the lattice. However, precisely because ηd is dimensionless (η has explicitly units of $1/d$, as the authors state), it has no units, so they should remove the arbitrary units label “a.u.” in both the main text and the Supplementary Information.

Our response:

We thank the Reviewer for the careful reading. We have removed “a.u.” in both the main text and the Supplementary Information.

Reviewer #5 commented:

2) Page 5, left column: “Assuming that $|E_n\rangle$ is the n -th eigenenergy of total Hamiltonian and ρ_{out} is the density matrix outside obtained by QST, then the probability of finding a particle of energy E_n outside the horizon can be obtained as”. If $|E_n\rangle$ lives in the 10-qubit dimensional Hilbert space and ρ_{out} is the reduced density matrix outside the horizon, living in a 7-qubit Hilbert space, how exactly is evaluated P_n ? Because the dimensions of the bracket do not agree between the density matrix and the states. This should be further clarified, if not in the text, at least in the Methods section.

Our response:

We thank the Reviewer for this suggestion. ρ_{out} is the 10-qubit density matrix reconstructed by the 7-qubit density matrix obtained by QST with the other three qubits being set to $|0\rangle$. We have clarified this in the “Calculation of radiation probability” of the Methods.

Reviewer #5 commented:

3) Page 5, right column: While referring to the experiments with shallow water waves in a tank, the authors state “In a much lower energy system like shallow water wave”. Well, I

would not say that a water tank at room temperature is a low energy system as compared to a condensate close to zero temperature... The authors should rewrite properly this sentence. Perhaps they mean in a “fully macroscopic setup” (since the Hawking temperature is determined by the scales of the setup; that is why they have such a low T_H)?

Our response:

We thank the Reviewer for this comment. The Hawking temperature T_H depends on the surface gravity g_h , which is related to the scales of the setup and determines the average energy of the radiated quasi-particles. Here “energy” actually refers to the average effective energy of radiated quasi-particles rather than the total energy of the system. For more clarity, we have rewritten this paragraph:

“This significant difference in magnitude is attributed to the scales of the setup in different experimental systems. In superconducting qubits, the coupling strength is usually on the order of MHz and thus the analogue surface gravity g_h is of the same magnitude, leading to $T_H = g_h/(2\pi) \sim 10^{-5}K$. Differently, the effective Hawking temperature of sonic black hole depends on the gradient of velocity at the analogue horizon. The BEC system and the shallow water wave system typically give us $T_H \sim 10^{-10} K$ [9–12] and $10^{-12}K$ [4], respectively.”

Reviewer #5 commented:

4) In Page 6, left column, it is not very clear why do they mean by “the entanglement between the pair in the black hole is protected by the analogue gravity so that it can slow the decrease of the concurrence”. This is a very misleading statement. Most likely, the authors mean that the Hawking radiation stimulated at the horizon prevents the loss of entanglement?

Our response:

Here we want to contrast the dynamics of concurrence for different spacetimes. To avoid misleading statements, we have rewritten that “The speed of entanglement propagation is limited by the gravitational effects near the horizon, and thus the decrease in concurrence is slowed in the curved spacetime case compared to the flat spacetime case.”

Reviewer #5 commented:

5) The authors work with two density matrices ρ_{in} , ρ_{out} to describe the interior and exterior of the black hole. However, they miss one qubit, the one at the horizon. Then, it is no longer true $S(\rho_{out}) = S(\rho_{in})$, even if the initial state is pure. While this is not critical, since $S(\rho'_{out}) = S(\rho_{in})$, with ρ'_{out} also containing Q_3 , the authors should correct the statement about $S(\rho_{out}) = S(\rho_{in})$ and leave more clear this aspect.

Our response:

We thank the Reviewer for the careful reading. We have corrected this in Methods and denoted $S(\rho_{in}) = S(\rho_{rest})$ as the entanglement between the interior of black hole and the rest part.

Reviewer #5 commented:

Last but not least, in general, I would further encourage the authors to polish the text of the manuscript.

Our response:

We thank the Reviewer again for the careful reading and useful comments on our work. We have polished the text of the manuscript based on the suggestions of the Reviewer. More importantly, the Reviewer clarified the nature of the Hawking radiation observed in our experiment. We are grateful for the professional help of the Reviewer.

Summary of changes

All our changes are marked in blue in the additional manuscripts. In the following we provide a summary of the main changes:

Main text:

1. We added “stimulated” in the abstract;
2. The arbitrary units label “a.u.” has been removed;
3. Page 4, right column, we deleted the sentence “Here we emphasize that the horizon appears after the initial state is ready. There are no additional excitations in the evolution” and added “Note that the Hawking radiation observed here is stimulated because we induce an excitation by flipping a qubit in $|1\rangle$ ”;
4. Page 5, left column, we deleted the sentence “It is also need to emphasize that we induce an excitation by flipping a qubit to $|1\rangle$ before the black hole appears, but the radiation is spontaneous after the black hole appears”;
5. Page 5, right column, we rewrote the sentence that “This significant difference is...”;
6. Page 6, left column, we added “the speed of entanglement propagation is limited by the gravitational effects near the horizon...”;
7. We added “Calculation of radiation probabilities” in Methods;
8. We corrected the statement in “Measure of entanglement” of Methods;

Supplementary Information:

1. The arbitrary units label “a.u.” has been removed;
2. We modified the statements in subsection “VII. E. Comparison of measured Hawking temperature with other experiments”.